# Sensitivity of the Ross Ice Shelf to environmental and glaciological controls

Francesca Baldacchino[1], Mathieu Morlighem[2,3], Nicholas R. Golledge[1], Huw Horgan[1], and Alena Malyarenko[4,1]

[1]Antarctic Research Centre,Victoria University of Wellington, New Zealand
[2]Department of Earth Sciences,Dartmouth College, Hanover, NH 03755, USA
[3]Department of Earth System Science, University of California Irvine, Irvine, CA 92697, USA
[4]National Institute of Water and Atmosphere Research, Wellington, New Zealand

**Correspondence:** Francesca Baldacchino (francesca.baldacchino@vuw.ac.nz)

**Abstract.** The Ross Ice Shelf (RIS) is currently stable but recent observations have indicated that basal melt rates beneath the ice shelf are expected to increase. It is important to know which areas of the RIS are more sensitive to enhanced basal melting as well as other external forcings or internal material properties of the ice to understand how climate change will influence RIS mass balance. In this paper, we use Automatic Differentiation and the Ice Sheet and Sea-level System Model to quantify the sensitivity of the RIS to changes in basal friction, ice rigidity, surface mass balance, and basal melting. Using Volume Above Flotation (VAF) as our quantity of interest, we find that the RIS is most sensitive to changes in basal friction and ice rigidity close to grounding lines and along shear margins of the Siple Coast Ice Streams and Transantarctic Mountains Outlet Glaciers. RIS sensitivity to surface mass balance is uniform over grounded ice, while the sensitivity to basal melting is more spatially variable. Changes in basal melting close to the grounding lines of the Siple Coast Ice Streams and Transantarctic Mountains Outlet Glaciers have a larger impact on the final VAF compared to elsewhere. Additionally, the pinning points and ice shelf shear margins are highly sensitive to changes in basal melt. Our sensitivity maps allow areas of greatest future vulnerability to be identified.

## 1 Introduction

Understanding and predicting how ice sheets will evolve in a warming world has become one of the most important questions for future climate change. An increase of 1.5 to 2°C in global temperature could lead to multi-meter rise in global sea-level over the next 100 to 1000 years (Raftery et al., 2017; Pattyn, 2018; Jenkins et al., 2018; Golledge et al., 2019). The Antarctic Ice Sheet (AIS) holds 91 percent of the global ice and consequently is the largest potential contributor to global sea level rise (Dirscherl et al., 2020; Mottram et al., 2020). AIS mass loss has accelerated over the past several decades primarily due to the intrusion of warm Circumpolar Deep Water (CDW) in ice shelf cavities in the Amundsen Sea Embayment, which has increased ice discharge into the ocean (Shepherd et al., 2012, 2018; Rignot et al., 2019).

Ice shelves along Antarctica's coast play a fundamental role in controlling mass flux because they buttress the outflow of mass by reducing longitudinal stresses at the grounding line (Schoof, 2007; Gudmundsson, 2013; Pattyn and Durand, 2013).

Recently, most of the mass loss from ice shelves has occurred through ocean-forced basal melting (Pritchard et al., 2012; Jenkins et al., 2018; Pattyn, 2018; Rignot et al., 2019; Robel et al., 2019). Ocean-forced basal melting reduces the buttressing effect of ice shelves allowing faster flow of grounded ice into the ice shelves. This causes greater discharge across the grounding line, and subsequent grounding line retreat (Moholdt et al., 2014; Pattyn et al., 2017; Shepherd et al., 2018; Gudmundsson et al., 2019).

The Ross Ice Shelf (RIS) is the largest cold-water ice shelf in Antarctica, buttressing both the West Antarctic Ice Sheet (WAIS) and the East Antarctic Ice Sheet (EAIS). These catchments, almost entirely buttressed by RIS, represent a total potential sea level rise contribution of 11.6m (Tinto et al., 2019). The stability of WAIS is a major uncertainty in predicting the AIS's future contribution to sea level rise (Pattyn and Durand, 2013; Jenkins et al., 2018). Recent research has indicated that a tipping point may have already been passed in some sectors of WAIS, causing irreversible grounding line retreat and thinning (Rignot et al., 2014; Golledge et al., 2019; Robel et al., 2019).

The RIS is currently stable, as its sub-ice shelf cavity has been insulated from inflows of warm CDW (Dinniman et al., 2011; Tinto et al., 2019; Das et al., 2020). Instead, basal mass loss is driven by subsurface inflows of cold, high salinity shelf water that melts ice near the grounding lines, and by seasonal inflows of summer-warmed Antarctic Surface Water (ASW) that melts shallow ice along the ice shelf front (Assmann et al., 2003; Stern et al., 2013; Stewart et al., 2019; Tinto et al., 2019; Adusumilli et al., 2020). These two different ocean processes cause basal melting, and thus mass loss on the RIS, to be variable in both space and time (Adusumilli et al., 2020). For example, basal melt rates exceeding 10 m a$^{-1}$ have been found at the grounding line of Byrd Glacier (Kenneally and Hughes, 2004), while annual-averaged rates on the order of 1 m a$^{-1}$ have been reported along the ice shelf front (Horgan et al., 2011; Moholdt et al., 2014; Stewart et al., 2019). Additionally, elevated basal melt rates of 10 m a$^{-1}$ on the ice shelf front near Ross Island have been observed in recent years due to intrusions of seasonal warm ASW (Stewart et al., 2019; Tinto et al., 2019; Das et al., 2020; Adusumilli et al., 2020). Previous modelling studies have found that ice shelf thinning close to Ross Island, could have a substantial impact on ice dynamics over a large region, including grounded ice up to 1000 km away at the WAIS grounding line of RIS (Fürst et al., 2016; Reese et al., 2018; Gudmundsson et al., 2019; Klein et al., 2020). However, averaged over the entire ice shelf, basal melt rates remain low, on the order of 0.1 m a$^{-1}$ (Rignot et al., 2013; Moholdt et al., 2014; Adusumilli et al., 2020) due to the low temperature of water masses on the Ross Sea continental shelf (Pritchard et al., 2012). Summer sea-ice concentrations in the Ross Sea are projected to decrease by 56% by 2050 (Smith Jr. et al., 2014). Ice-free period is also expected to increase (Dinniman et al., 2018), which will very likely increase ice-shelf basal melting and, subsequently, affect the future stability of the RIS (Stewart et al., 2019).

It remains unclear whether changes in external forcings (such as basal melting and surface mass balance) or other processes (such as the internal material properties of the ice) could cause more mass loss in the future. Moreover, glaciers and ice sheets can respond differently to different melt rate distributions even when the total integrated melt is the same (Gagliardini et al., 2010). Hence, the spatial pattern of melt rates has a stronger impact than the area-averaged value of basal melt. In addition, it has been shown that basal melting at the grounding line causes increased grounding line retreat, indicating that knowledge of basal melt distribution is critical for accurate prediction of grounding line migration (Walker et al., 2008).

This paper investigates which changes in external forcings and internal material properties of the ice affect the overall mass balance of the system and identifies areas of sensitivity on the RIS. The external forcings investigated are ocean-forced melt rates and surface mass balance. The internal material properties we investigate here are ice rigidity and basal friction. These four properties are chosen as they each have the potential to enhance ice discharge and Volume Above Flotation (VAF) of the RIS (Shepherd et al., 2012, 2018; Rignot et al., 2019). These parameters are explored using the Automatic Differentiation (AD, M. Sagebaum, 2019) tool in the Ice-sheet and Sea-level System Model (ISSM). The AD tool produces a spatial map identifying where, and how sensitive, VAF is to parameter changes. VAF is calculated for the entire domain, while the sensitivity is specific to the location concerned.

## 2  Methods

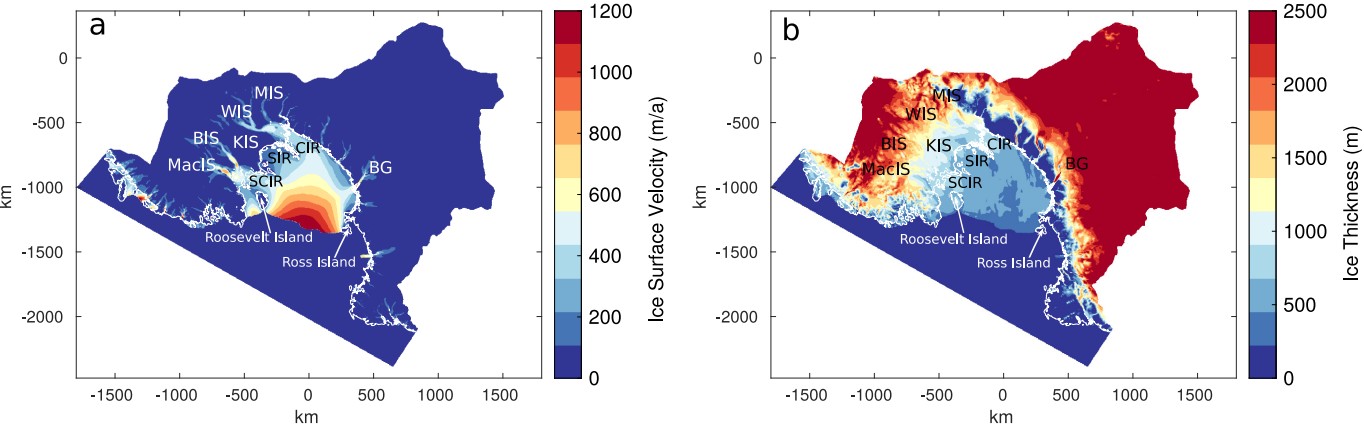

**Figure 1.** Modelled Ross Ice Shelf surface velocities (a) and ice thickness (b) after initialisation. The grounding line is marked in white. Locations discussed in this paper are labelled. These include the Siple Coast Ice Streams: Mercer Ice Stream (MIS), Whillans Ice Stream (WIS), Kamb Ice Stream (KIS), Bindschadler Ice Stream (BIS) and MacAyeal Ice Stream (MacIS). Byrd Glacier (BG) and Ross Island are also labelled. In addition, the ice rises are labeled on the Siple Coast: CIR = Crary Ice Rise, SIR = Steershead Ice Rise, SCIR = Shirase Coast Ice Rumples and Roosevelt Island. The projection of this map and all others presented is polar sterographic with a true scale at -71° (EPSG:3031).

We use the Ice-sheet and Sea-level System Model (ISSM) to explore the influence that changes in the chosen parameters have on the mass balance of the model domain. We use a Shallow Shelf approximation (SSA, (MacAyeal, 1989)) with a rheology based on Glen's flow law (Glen, 1955) and ice viscosity depending on the ice temperature (Cuffey and Paterson, 2010) (Figures A1 and A2). We use the ice temperature from ISSM's submission to the Ice Sheet Model Intercomparison Project for CMIP6 (ISMIP6) to initialize the ice viscosity (Seroussi et al., 2020). The basal friction is based on a Budd friction law (Budd et al., 1979), in which basal drag is directly proportional to sliding velocity. This friction law may not be valid under some sectors

of our model domain such as the Siple Coast. Therefore, we performed additional experiments to test the sensitivity of our results to the Budd friction law (Figure A3) by using a Weertman friction law (with an exponent of $m = 3$) (Weertman, 1957) instead. The grounding line evolves assuming hydrostatic equilibrium following a sub-element grid scheme (i.e. Sub-element Parameterization 1 in Seroussi et al. (2014)) and the ice front remains fixed through time during all the simulations performed.

The model domain covers the entire RIS and its tributaries as observed today (Zwally et al., 2012) (Figure 1a). Ice thickness and bed elevation are interpolated from the BedMachine v2 dataset (Morlighem and Binder, 2020) (Figure 1b). Our simulations rely on a non-uniform mesh that has a resolution of 1 km at the grounding lines and in the shear margins, increasing to 20 km in the ice sheet interior, and with a nominal resolution of 10 km within the ice shelf. The basal friction coefficient is inferred through a data assimilation technique (Morlighem et al., 2013) to reproduce observed InSAR surface velocities from the MEaSURES data-set (Rignot et al., 2017). Environmental boundary conditions include RACMO2.3p2 surface mass balance (Van Wessem et al., 2018) and basal melt rates which are calculated using the MITgcm ocean-ice shelf model (Losch, 2008; Holland and Jenkins, 1999; Davis and Nicholls, 2019). The ocean model was initialised and forced with ECCO2 reanalysis (Menemenlis et al., 2005) for 2006-2016 and has compared well with observed seasonal warm water inflow into the cavity and high summer melt rates in the frontal RIS in 2011-2014 (Malyarenko et al., 2019; Stewart et al., 2019). The ice sheet model is run forward for 4 years to allow the grounding line position and ice geometry to relax. These results define the initial state of our control run and sensitivity experiments (Figure 1).

A 20 year forward simulation forced by Surface Mass Balance (SMB) from RACMO2.3p2 1979-2014 mean (Van Wessem et al., 2018) and by ice shelf basal melt rates from MITgcm 2006-2016 weekly outputs is used in the Automatic Differentiation (AD) package within ISSM. The AD tool maps the sensitivity of ice Volume Above Flotation (VAF) after a 40-year simulation to changes in the chosen parameter (i.e. ice rigidity, basal friction, SMB and basal melt) on the RIS domain. We follow a similar approach as the one described in Morlighem et al. (2021). Specifically, we investigate the sensitivity of the model with respect to the ice rigidity parameter, $B$, which controls the ice viscosity defined as:

$$\mu = \frac{B}{2\,\dot{\varepsilon}_e^{\frac{n-1}{n}}} \tag{1}$$

where $\mu$ is the ice viscosity, $\dot{\varepsilon}_e$ is the effective strain rate, and $n = 3$ is Glen's exponent. We also consider the sensitivity of the model's volume above flotation to the basal friction coefficient parameter, $C_b$, defined as:

$$\tau_b = C_b{}^2 N\, v_b \tag{2}$$

where $\tau_b$ is the basal stress, $N$ is the effective pressure (assuming perfect hydrological connectivity to the ocean), and $v_b$ is the sliding velocity. Depth-integrated mass continuity arises from:

$$\frac{\partial H}{\partial t} = \nabla \cdot H \bar{\mathbf{v}} + \dot{M}_s - \dot{M}_b \tag{3}$$

where $H$ is the ice thickness, $\bar{\mathbf{v}}$ is the depth integrated ice velocity, $\dot{M}_s$ is the surface mass balance and $\dot{M}_b$ is the ocean-induced melt under floating ice.

Automatic differentiation provides the gradient of the final VAF, $V$, to a model parameter $P$: $\mathcal{D}V(P)$. In other words, the first order response of the VAF to a given perturbation $\epsilon \delta P$ in $P$ (where $\epsilon \in \mathbb{R}$, and $\delta P$ is a field defined over the entire model domain $\Omega$ that can be spatially variable) is given by:

$$V(P + \epsilon \delta P) = V(P) + \epsilon \int_\Omega \mathcal{D}V(P)\, \delta P\, d\Omega + \mathcal{O}(\epsilon^2). \tag{4}$$

The gradient, $\mathcal{D}V(P)$, therefore highlights the regions where the model is most sensitive to changes in $P$, and the regions where changes in $P$ would not affect the final VAF at a first order.

## 3   Results

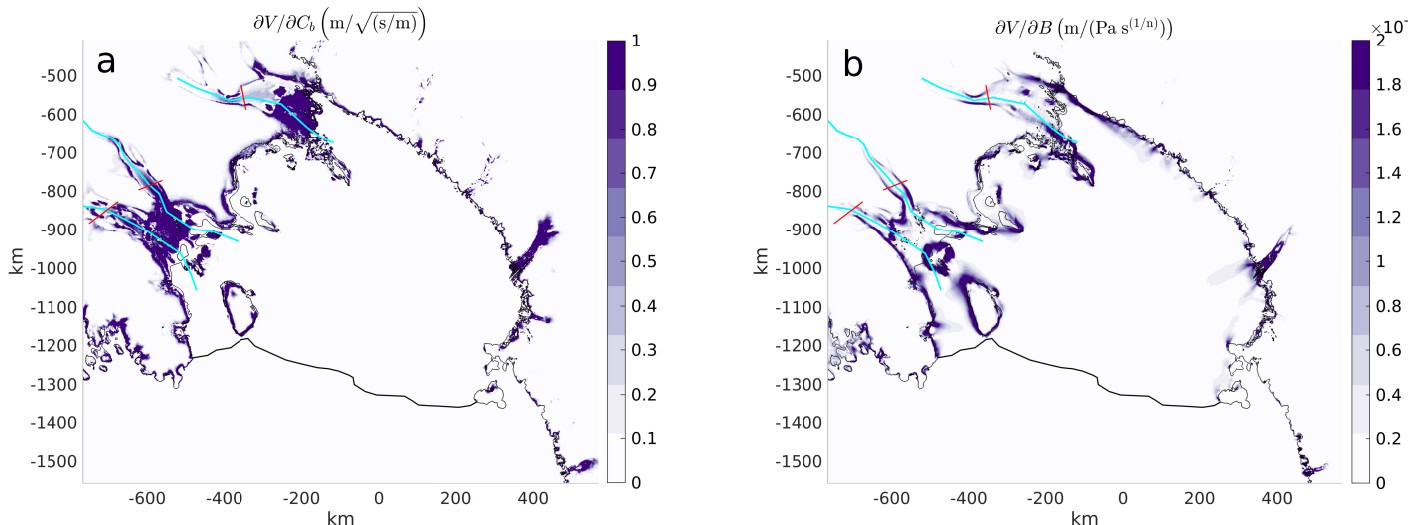

**Figure 2.** Sensitivity maps of final volume above flotation to the basal friction coefficient $C_b$ (a) and ice rigidity $B$ (b) over 40 years. The blue lines highlight the tracks for the along-flow profiles and the red lines the across-flow profiles.

Figure 2 shows the sensitivity maps of the model to the basal friction coefficient (Figure 2a), and the ice rigidity parameter (Figure 2b). These maps show that the sensitivity to basal friction and ice rigidity is low in the majority of the grounded domain, which means that changing the basal friction or ice rigidity over the majority of the region would not significantly influence the overall mass balance over 40 years. Conversely, sensitive areas include the vicinity of the grounding lines, with the active Siple Coast Ice Streams (Mercer, Whillans, Bindschadler and MacAyeal) and Byrd Glacier being the most noticeable grounding lines in the domain. There is high sensitivity to basal friction and ice rigidity along and inboard (i.e. inside of the shear margins) of the ice streams' margins which continues for more than 200 km upstream of the grounding lines. Figure 3 shows that the sensitivity to basal friction and ice rigidity increases at the major Siple Coast Ice Streams' grounding lines with the sensitivity

to ice rigidity continuing to increase downstream of these grounding lines. Negative values are observed in Figure 3, which are likely to be numerical artefacts and should be interpreted with caution (Morlighem et al., 2021). Figure 4 shows that basal

friction and ice rigidity sensitivities are highest at the Whillans (Figure 4a,d) and MacAyeal Ice Stream margins (Figure 4c,f). Additionally, Figure 4 shows that basal friction sensitivities are highest within Bindschadler Ice Stream trunk (Figure 4b) and ice rigidity sensitivities are highest at its margins (Figure 4e). These sensitivities are positive, highlighting that an increase in basal friction or ice rigidity in these areas would cause an increase in the overall VAF, and also that a decrease in basal friction or ice rigidity would induce a decrease in VAF.

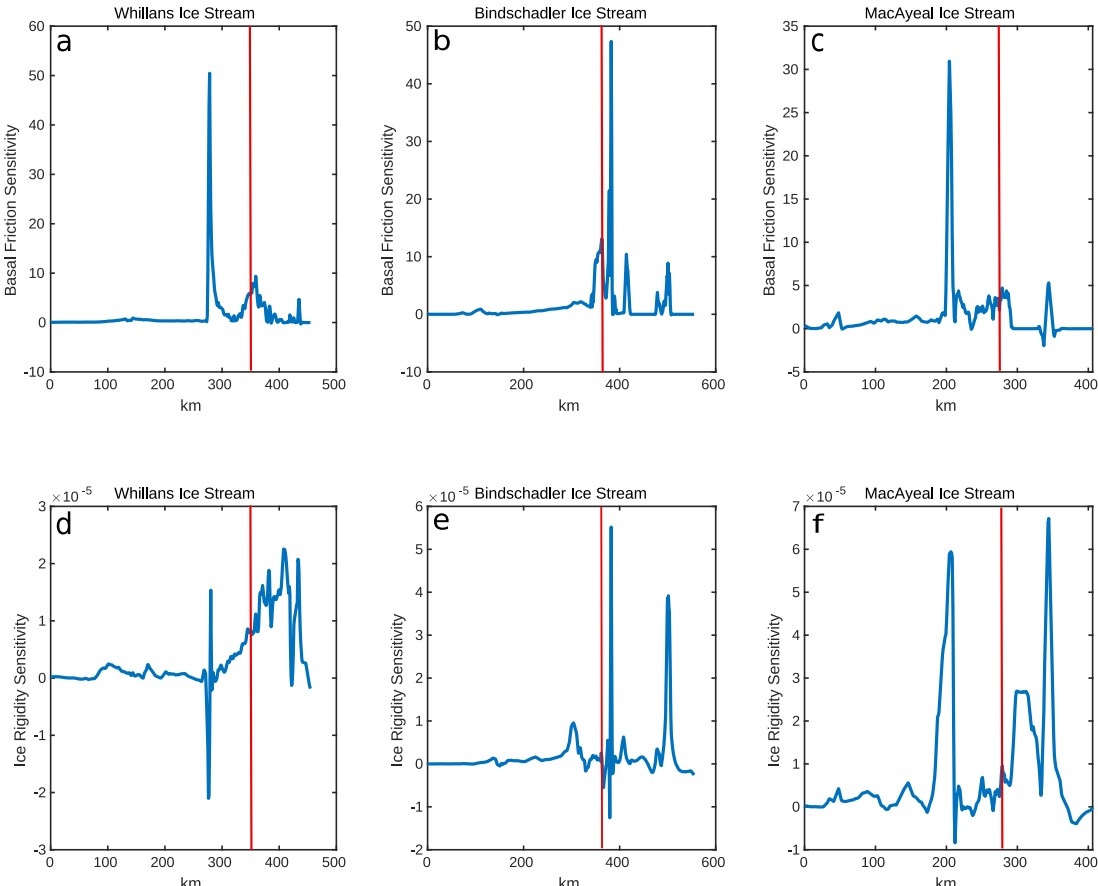

**Figure 3.** Along-flow profiles of the major active Siple Coast Ice Streams' sensitivity to glaciological controls: basal friction ($m/\sqrt{(s/m)}$) and ice rigidity ($m/(Pa\ s^{(1/n)})$). X axis (distance) increases in the downflow direction. The vertical red line highlights the position of the grounding line.

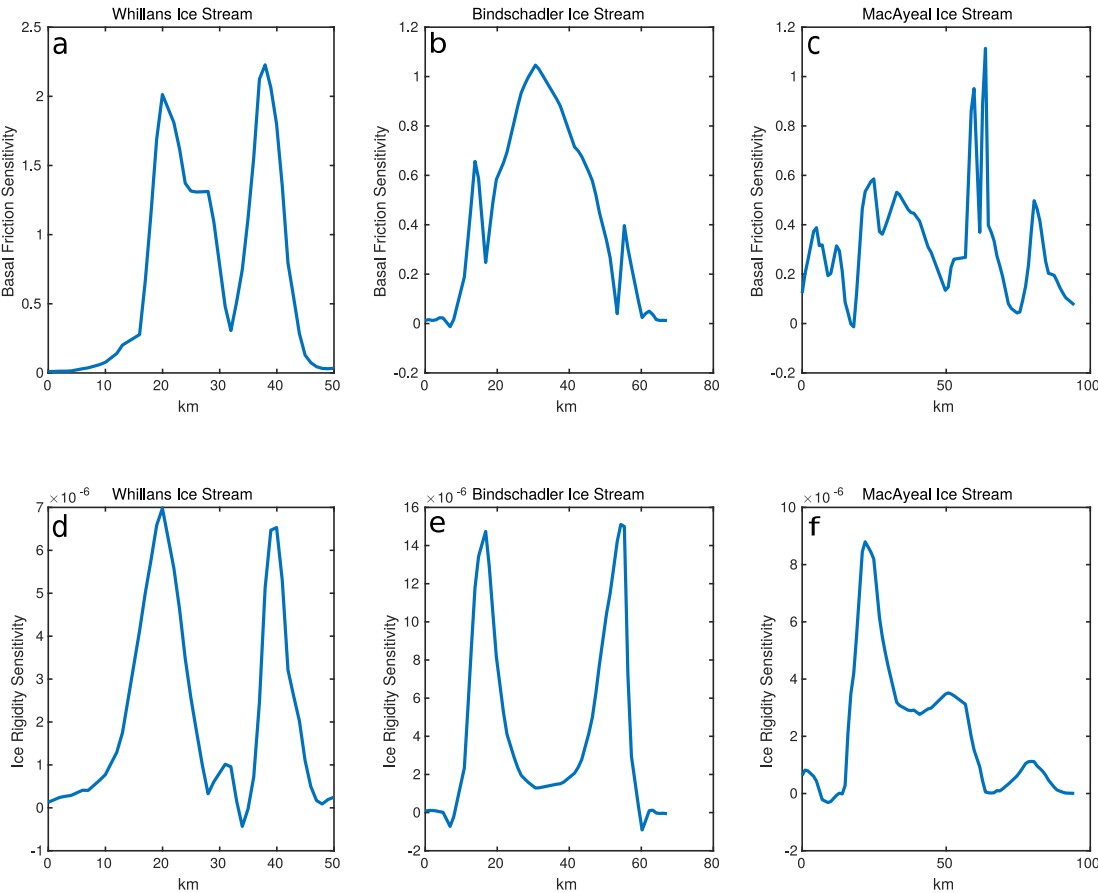

**Figure 4.** Across-flow profiles of the major active Siple Coast Ice Streams' sensitivity to glaciological controls: basal friction ($m/\sqrt{(s/m)}$) and ice rigidity ($m/(Pa\ s^{(1/n)})$). These across-flow profiles display sensitivities across the ice streams including their margins.

Kamb Ice Stream on the Siple Coast shows low to no sensitivity to changes in the basal friction or ice rigidity, highlighting that the Kamb Ice Stream is currently stagnant and thus changing the friction or ice rigidity will not change the ice discharge significantly. The sensitivity map tests the sensitivity of the VAF to perturbations in the forcing parameters, assuming that the general ice sheet configuration does not change. Therefore, the sensitivity map is unable to capture the general reactivation of the Kamb Ice Stream as the reactivation of a dormant ice stream would be a large perturbation that would constitute a change

in the general ice sheet configuration. In addition, we see that the pinning points of the RIS such as Roosevelt Island and Ross Island are sensitive to changes in basal friction and ice rigidity. In particular, we see that the smaller ice rises (i.e. Crary Ice Rise and Steershead Ice Rise) downstream of the Siple Coast Ice Streams show high sensitivity to changes in ice rigidity. Finally, the

interior of the ice sheet, where the ice is slower, shows lower sensitivities due to the time interval used in the model simulation (Morlighem et al., 2021). The sensitivity maps show changes in basal friction and ice rigidity over 40 years, therefore it will

take some time before changes in velocities upstream impact ice discharge, ice thickness and consequently mass balance of the RIS.

Figure 5 shows the sensitivity maps of the model with respect to changes in external forcings: surface mass balance, $\dot{M}_s$ (Figure 5a), and basal melting, $\dot{M}_b$ (Figure 5b) over a 40 year period. The sensitivity map related to SMB (Figure 5a) shows that most of the grounded ice has a clear positive sensitivity that gradually decreases to 0 as the floating ice is reached. This is

expected since floating ice does not contribute directly to VAF, whereas increasing surface mass balance over the grounded ice would lead to a direct increase in VAF. Figure 6 highlights the reduction of sensitivity to SMB that occurs across the grounding zones of the major Siple Coast Ice Streams'. The high spikes observed in Figure 6 downstream of the grounding lines (at 500km for Bindschadler Ice Stream (Figure 6b) and 350km for MacAyeal Ice Stream (Figure 6c)) are numerical artefacts and should be interpreted with caution. These numerical artefacts are likely due to the sub-element parameterization used within

the numerical model's treatment of the grounding line (Morlighem et al., 2021). The transition zone between high sensitivity and low sensitivity is larger in regions of fast flow such as the Siple Coast Ice Streams and Byrd Glacier compared to regions of slower flow (Figure 5a). Figure5a further demonstrates that the Roosevelt Island and Ross Island pinning points are sensitive to changes in SMB as are the smaller ice rises downstream of the Siple Coast Ice Streams (i.e. Crary Ice Rise and Steershead Ice Rise).

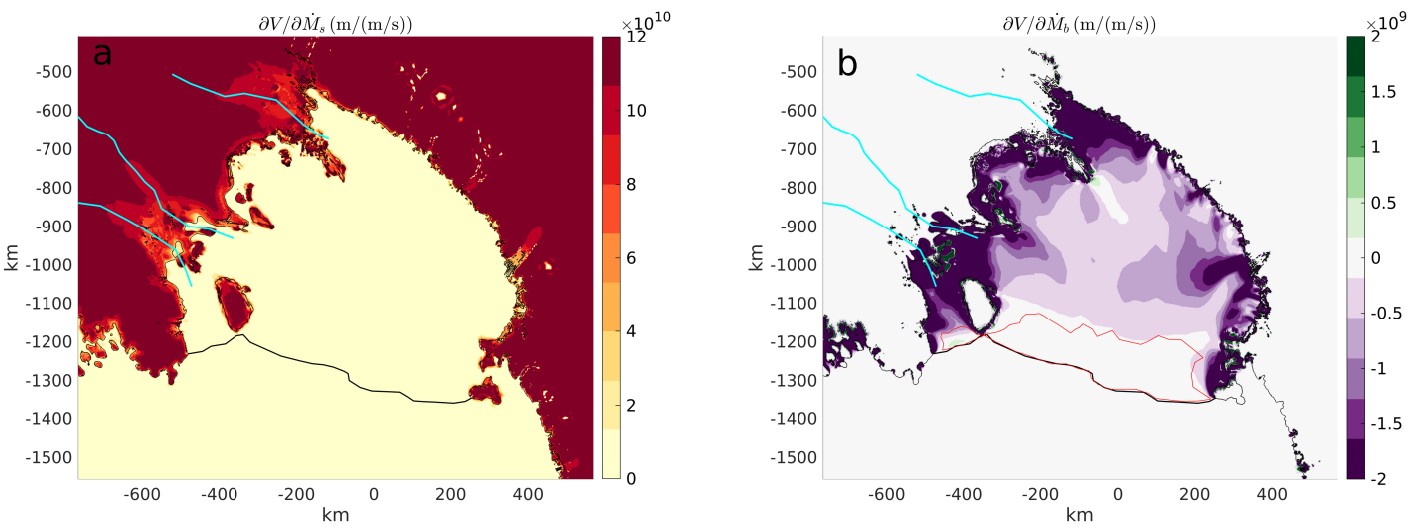

**Figure 5.** Sensitivity maps of final volume above flotation to the surface mass balance $\dot{M}_s$ (a) and basal melting $\dot{M}_b$ (b) over 40 years. The passive ice on the RIS identified by Fürst et al. (2016) is outlined in red. The blue lines highlight the tracks for the along-flow profiles.

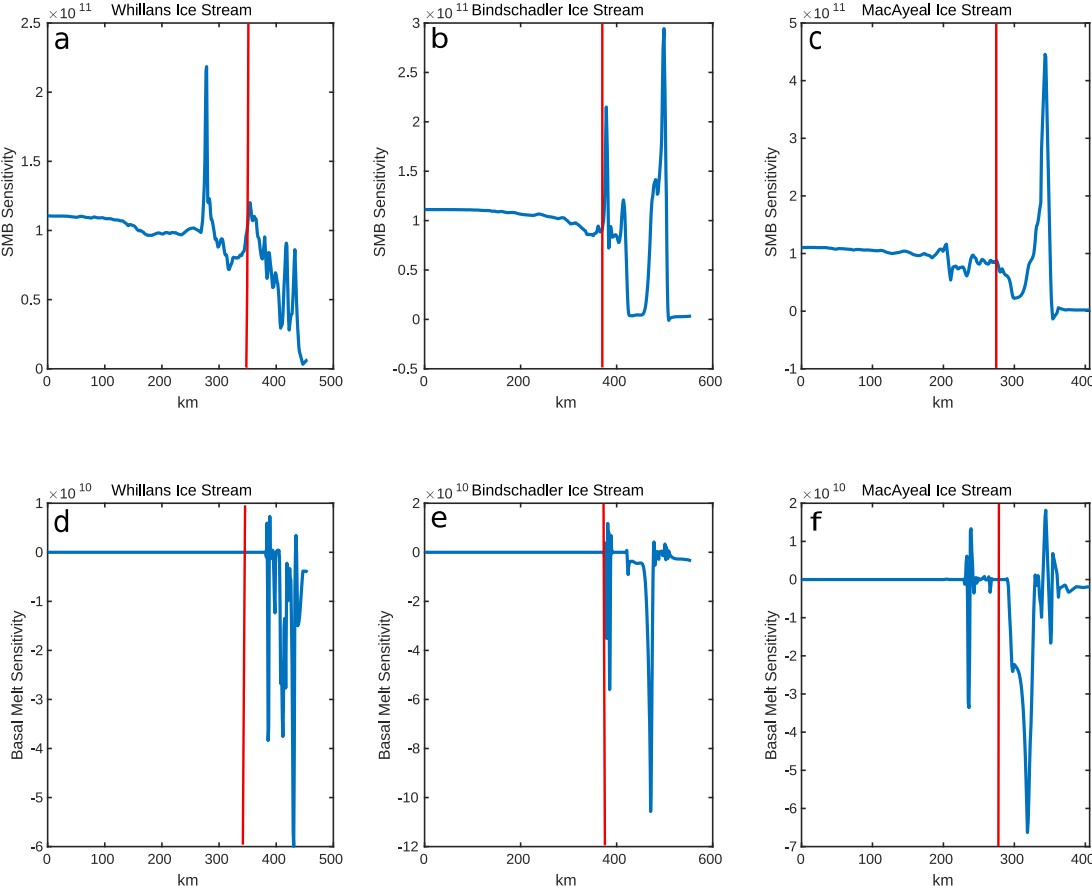

**Figure 6.** Along-flow profiles of the major active Siple Coast Ice Streams' sensitivity to environmental controls: surface mass balance ($m/(m/s)$) and basal melting ($m/(m/s)$). X axis (distance) increases in the down-flow direction. The vertical red line highlights the position of the grounding line.

Finally, the sensitivity to basal melting is zero over grounded ice (Figures 5b and 6) as expected because the model only allows basal melting to be applied on elements that are fully floating, but there is high spatial variability on the floating ice. Figure 5b shows a consistently high sensitivity at all grounding lines of the RIS and along the western and eastern ice-shelf shear zones over a 40 year period. In particular, we observe a strong sensitivity at the Siple Coast Ice Streams' and Byrd Glacier's grounding lines (including the vicinity of the grounding lines), and the Ross Island shear zone. Additionally, areas of the ice shelf directly downstream from these active ice streams and outlet glaciers are highlighted as sensitive to basal melting changes. The Kamb Ice Stream has high sensitivity directly at the grounding line and near Crary Ice Rise and Siple Dome, while the Bindschadler and MacAyeal Ice Streams have a broad range of sensitivity downstream all the way to Roosevelt

Island. All these sensitivities are negative, meaning that an increase in basal melt in these regions would lead to a decrease in the final VAF. Figure 6 confirms that the sensitivity to changes in basal melting is highest in the vicinity of the grounding line

of the major Siple Coast Ice Streams and then reaches negligible sensitivity on the floating ice shelf. In particular, Figure 6 shows that Bindschadler Ice Stream experiences the highest sensitivities in the vicinity of grounding line (Figure 6e) compared to Whillans (Figure 6d) and MacAyeal Ice Streams (Figure 6f). In addition, Figure 5b shows high sensitivities at the pinning points across the RIS: Roosevelt Island, Crary Ice Rise, Steershead Ice Rise and the Shirase Coast Ice Rumples. Finally, no sensitivity to changes in basal melting are observed near the calving front of the RIS (i.e. red outline in Figure 5b showing the

passive ice region identified by Fürst et al. (2016)).

## 4    Discussion

### 4.1    Sensitivity to changes in basal friction and ice rigidity

The model shows that the vicinity of the grounding lines of the active Siple Coast Ice Streams are particularly sensitive to changes in basal friction over a 40 year period. Previous studies have shown that the Siple Coast Ice Streams are underlain by

deformable till that is water-saturated (Kamb, 1991; MacAyeal, 1992; Joughin et al., 2004). Despite this deformable substrate, basal friction at the beds and along the lateral margins are the primary forces resisting flow in these ice streams (Ranganathan et al., 2021). Spatially varying basal friction has been observed at the grounding lines of Bindschadler and MacAyeal Ice Streams with localised areas of relative high drag or subglacial 'sticky spots' providing large resistances to ice flow (MacAyeal, 1992; Joughin et al., 2004; Ranganathan et al., 2021). Our results show that the Bindschadler (Figure 3b) and MacAyeal Ice

Streams (Figure 3c) have high sensitivities to changes in basal friction at their grounding lines due to these 'sticky spots' playing an important role in controlling the ice discharge. Additionally, our results show that the Whillans Ice Stream has high sensitivities to basal friction changes at its grounding line (Figure 3a). Horgan et al. (2021) found relatively stiff till at the grounding line of Whillans Ice Stream compared to the main ice trunk. The stiffer till at the grounding line creates more basal drag and consequently a slowdown of ice, leading to a thickening of ice which stabilizes the position of the grounding line

(Joughin et al., 2005; Anandakrishnan et al., 2007; Horgan et al., 2021). A softening of the till at the grounding line through basal friction changes would change the position of the grounding line and discharge rates. This agrees with our results that the vicinity of the grounding lines of Whillans, Bindschadler and MacAyeal Ice Streams display high sensitivities to changes in basal friction as their ice discharge rates are largely controlled by the basal friction conditions. We also see in our results that the Siple Coast Ice Streams' shear margins are highly sensitive to changes in basal friction. Winberry et al. (2009) found that

the MacAyeal Ice Stream has a stick-slip behaviour, suggesting that the basal friction within the main ice trunk is important in controlling the ice discharge, which agrees with our results. In addition, it has been suggested that internal heating interacts with drainage at the bed forming a channelised drainage system (Suckale et al., 2014; Ranganathan et al., 2021). Our results show that the Bindschadler Ice Stream margins and main ice trunk are sensitive regions to changes in basal friction (Figure 4b). Meyer and Minchew (2018) found evidence of a channelised drainage system near the temperate zones of Bindschadler

Ice Stream shear margins, which has the potential to change the basal friction and cause changes in the ice stream discharge

and flow rates. Additionally, Van Der Veen et al. (2007) found lubricated conditions under Whillans Ice Stream due to nearby meltwater production under its shear margins. Our results show that Whillans Ice Stream is highly sensitive to changes in basal friction at its shear margins (Figure 4a), suggesting that changes in lubrication conditions here influence the flow and discharge rates of the ice stream.

In addition, our results show that the Siple Coast Ice Streams are highly sensitive to changes in ice rigidity over a 40 year period at their grounding lines. Meyer and Minchew (2018) show that increasing ice softness at ice stream margins leads to enhanced ice discharge. Our results also show that changes in ice rigidity at the grounding lines (including in the vicinity of the grounding lines) of the Siple Coast Ice Streams affect the ice flow and discharge rates impacting the overall mass balance of the RIS domain. We also see in our results that the Siple Coast Ice Streams' shear margins are highly sensitive to changes in ice rigidity. Previous studies have found that temperate ice has formed in the Siple Coast Ice Stream shear margins due to internal heating resulting from changes in crystalline structures (Schoof, 2004; Ranganathan et al., 2021). Minchew et al. (2018) found that the development of crystalline fabric is an important control on ice rigidity in the shear margins, with ice becoming progressively softer downstream due to shear heating. Ice stream widening has been related to this formation of temperate ice and as a consequence a reduction in ice rigidity (Schoof and Hewitt, 2013; Hunter et al., 2021). Our results show that Bindschadler Ice Stream has the highest sensitivity to changes in ice rigidity at the margins (Figure 4e), while MacAyeal Ice Stream has a higher sensitivity in the main ice trunk (Figure 4f). Meyer and Minchew (2018) show that Bindschadler Ice Stream has zones of temperate ice within its shear margins, with these zones being at melting temperature and controlling the rate at which the ice stream flows by lowering the ice rigidity. Therefore, changes in the ice rigidity of the Bindschalder Ice Stream margins would influence the ice stream discharge as shown by our results. Recently, the Whillans and MacAyeal Ice Streams have been decelerating, while the Bindschadler Ice Stream has been accelerating due to changes at the ice streams' beds and margins (Hulbe and Fahnestock, 2004; Joughin et al., 2005; Van Der Wel et al., 2013) highlighting that changes in basal friction and ice rigidity have a large influence over ice stream discharge and thus mass balance.

Our results also show that changes in basal friction and ice rigidity at Byrd Glacier would lead to large changes in the rate of mass loss over a 40 year period. Byrd glacier is the largest outlet glacier draining through the Transantarctic Mountains and it has been shown that Byrd Glacier experiences variability on a range of timescales as a response to changes in its subglacial hydrogical conditions (Stearns et al., 2008). This concurs with our results showing that Byrd Glacier is highly sensitive to changes in basal friction within its main ice trunk and shear margins. In addition, it has been shown that fast deformation persist in the Byrd Glacier shear margins (Ledoux et al., 2017) and this makes it highly sensitive to changes in ice rigidity, which is also shown by our results.

Additionally, high sensitivity to changes in basal friction and ice rigidity is found near pinning points (i.e. Ross Island, Roosevelt Island, Steershead and Crary ice rises and Shirase Coast Ice Rumples) on the RIS. These regions are important for the stability of the RIS because they provide resistance to the ice flow from grounded ice and help buttress the ice shelf (Dupond and Alley, 2005; Fürst et al., 2016; Reese et al., 2018; Still et al., 2019). Therefore, a change in the basal friction and ice rigidity in these regions would change the resistive effect of pinning points on the RIS (Meyer and Minchew, 2018; Alley et al., 2019; Still et al., 2019) and overall mass balance as shown by our results.

## 4.2 Sensitivity to changes in surface mass balance

The sensitivity map for SMB changes (Figure 5a) shows that the longer the ice remains grounded the larger the sensitivity to changes in SMB. Our results show that grounded ice is highly sensitive to changes in SMB while floating ice has near zero sensitivity. Sensitivity to changes in SMB is observed in the vicinity of the grounding lines, with the Whillans Ice Stream grounding line showing high sensitivity over a 40 year period (Figure 6a). Previous studies have shown that Whillans Ice Stream has thicker ice at the grounding line, which stabilizes the grounding line position (Joughin et al., 2005; Anandakrishnan et al., 2007). Therefore, our results show high sensitivity at the Whillans Ice Stream grounding line due to changes in SMB driving changes in the ice thickness, grounding line stability and ice stream discharge. In addition, high sensitivity to changes in SMB is found at the Ross Island and Roosevelt Island pinning points on the floating ice shelf. Pinning points are localised areas of grounding within a floating ice shelf and thus any changes in the SMB here would affect ice thickness and discharge changes on the RIS (Reese et al., 2018; Gudmundsson et al., 2019; Still et al., 2019).

## 4.3 Sensitivity to changes in basal melting

Furthermore, our results show that the grounding lines of the Siple Coast Ice Streams are highly sensitive to changes in basal melting over a 40 year period. The grounding lines (including the vicinity of the grounding lines) show high negative sensitivities because changes in basal melting in these regions lead to changes in grounding line position, ice discharge and thus mass loss (Moholdt et al., 2014; Pattyn et al., 2017; Shepherd et al., 2018; Gudmundsson et al., 2019). Basal melting can influence changes in basal friction and ice rigidity causing the Siple Coast Ice Streams' dynamics to change. For example, seasonal and interannual variations in ocean heat content will dictate the temperature gradient of the ice at the grounding line and rate of basal melting, changing basal friction and ice rigidity. Basal melt is applied as a forcing with no feedback in the numerical model resulting in the sensitivity maps not taking into account the effect of ocean heat or rate of basal melting but only changes in basal melt. Currently, the RIS has generally low basal melt rates near the grounding lines due to the relatively shallow and uniform ice draft as well as weaker tidal currents (Moholdt et al., 2014). However, Marsh et al. (2016) found changes in basal melt rates of 15-22 m a$^{-1}$ near the grounding line of the Whillans Ice Stream due to high subglacial discharge forming a well-defined basal channel. In addition, Adusumilli et al. (2020) found varying basal melt rates of 0-1.5 m a$^{-1}$ at the grounding lines of the active Siple Coast Ice Streams. This highlights that changes in basal melting are currently occurring in sensitive areas that have the greatest impact on the final VAF compared to elsewhere.

Our results also show high negative sensitivities to changes in basal melting at the grounding line of Byrd Glacier. These high negative sensitivities are due to Byrd Glacier having a reverse bedrock slope, which drives changes in grounding line position and ice discharge in response to changes in basal melting (Schoof, 2007; Stearns et al., 2008; Van Der Veen et al., 2014; Pattyn et al., 2017). Adusumilli et al. (2020) found basal melt rates ranging 4-6 m a$^{-1}$ on interannual timescales at the grounding line of Byrd Glacier due to inflows of cold, High Salinity Shelf Water indicating that this region is important to monitor in relation to the future mass balance of the RIS. We see that the active Siple Coast Ice Streams and Byrd Glacier have similar sensitivities to changes in basal friction, ice rigidity and basal melting. However, changes at the Siple Coast Ice Streams

result in far greater ice loss as they discharge large volumes of ice directly from WAIS into the RIS, they have deformable till at their beds and their shear margins can migrate (Shabtaie and Bentley, 1987; Bennett, 2003; Bindschadler et al., 2003; Catania et al., 2012; Muto et al., 2013). However, the numerical model is only able to capture the lower bound sensitivity estimates of the Siple Coast Ice Streams as the model does not include these processes which would favour greater ice discharge and mass loss.

In addition, high sensitivity to changes in basal melt is also found near pinning points (i.e. Ross Island, Roosevelt Island, Steershead and Crary ice rises and Shirase Coast Ice Rumples) on the RIS. Changes in basal melting can result in detachment from pinning points and grounding line retreat (Still et al., 2019). Once the ice shelf has become detached from these pinning points, the balance of forces across the RIS will change inducing changes in ice flow, thickness and thus mass balance (Still et al., 2019; Reese et al., 2018). Gudmundsson et al. (2019) indicated that rapid melting identified near Ross Island influences a structurally critical region in which ice thickness changes can influence the flow speed of the entire RIS. Additionally, Reese et al. (2018) highlighted that the Bindschadler Ice Stream flow was influenced in response to thinning at the Ross Island region (more than 900 km away). Our results concur with these studies in that Ross Island is an important region to monitor due to it being identified as a region of high buttressing potential (Fürst et al., 2016) and its high sensitivity to changes in basal melting having a large impact on the final mass balance of the RIS.

Finally, the calving front of the RIS shows no sensitivities to changes in basal melting impacting the final VAF over a 40 year period. This is consistent with Fürst et al. (2016), who identified this region as "passive". However, removal of this 'passive' region could result in the current summer production of the warm ASW (Stewart et al., 2019) reaching sensitive areas of the RIS (Fürst et al., 2016). Tinto et al. (2019) show that the role of climate variations in destabilizing the RIS may have depended on the position of the ice shelf front in the past. For example, during the last glacial maximum, the ice sheet grounding line was near the edge of the continental shelf and thus relatively warm CDW would have been able to flow into the sub-ice cavity, generating high melt rates at the grounding lines of the RIS (Tinto et al., 2019). We see that there is higher negative sensitivities to changes in basal melting at the grounding lines compared to the ice shelf shear zones and pinning points of the RIS. However, there needs to be a change in the position of the ice shelf front, ocean circulation/mixing and strength of tidal currents to encourage changes in basal melt rates at the grounding lines of the RIS (Moholdt et al., 2014).

## 4.4 Limitations

Our model relies on a Budd linear sliding law, in which the basal drag coefficient ($\sqrt{(s/m)}$) is directly proportional to sliding velocity. This friction law may not be valid under some sectors of our model domain such as the Siple Coast. To investigate this possibility we performed additional experiments to test the sensitivity of our results to the friction law and length of transient simulation (Figure A3) by using a Weertman friction law (Weertman, 1957) with a sliding exponent of $m = 3$. The sensitivity of the model's volume above flotation to the basal friction coefficient parameter, $C_w$, is defined as:

$$\tau_b = C_w{}^2 v_b^{1/m} \tag{5}$$

where $\tau_b$ is the basal stress and $v_b$ is the sliding velocity. In line with Morlighem et al. (2021), we find that the results of the Weertman friction law are quantitatively similar to that of the Budd sliding law, suggesting that the areas highlighted in the sensitivity maps are robust features. Extending the length of the transient simulation increases the sensitivities accordingly (i.e. doubling the simulation time will double the sensitivity) but the sensitivity patterns remain qualitatively similar. These results suggest that the conclusions drawn from the sensitivity maps are reliable and related to the physics of the system not the model parameters (Morlighem et al., 2021).

## 5 Conclusions

In this study, we demonstrate the capability of AD to provide fine scale sensitivity maps of the RIS domain. These sensitivity maps indicate where changes in the environmental and glaciological controls would have the greatest impact on the mass balance of the RIS domain over a 40 year period. Overall, we find that the final VAF for the modelled domain is highly sensitive to changes in basal friction and ice rigidity at the grounding lines and shear margins of the active Siple Coast Ice Streams and Byrd Glacier. The sensitivity to changes in these glaciological controls at the RIS pinning points is smaller compared to the grounding lines. However, these pinning point regions have potentially larger dynamical impacts across the RIS due to being regions of high buttressing potential. We also find that the final VAF for the modelled domain is highly sensitive to changes in SMB over the grounded portion of the RIS, and also highly sensitive to changes in basal melting at the grounding lines, pinning points and ice shelf-shear margins of the RIS.

The highest sensitivity to changes in both glaciological and environmental controls is observed at the grounding lines of the Siple Coast Ice Streams and Transantarctic Mountains Outlet Glaciers. These results show that the RIS mass balance is highly sensitive to changes in ocean circulation and mixing as well as tidal currents which could potentially drive changes in basal melt rates at the calving front and at the grounding lines. More work is needed to understand the complex ice-ocean interactions at the groundings lines. Thus, these sensitivity maps should be used to inform future field campaigns as to which areas of the RIS are sensitive and should be monitored.

*Data availability.* The Ice-sheet and Sea-level System Model can be accessed at https://issm.jpl.nasa.gov (we used version 4.18). BedMachine Antarctica is available at NSIDC (http://nsidc.org/data/nsidc-0756). InSAR-Based ice velocity is found at NSIDC (https://nsidc.org/data/nsidc-0484/). The Antarctic surface mass balance (RACMO 2.3p2) is available at https://www.projects.science.uu.nl/iceclimate/models/racmo-data.php

## Appendix A

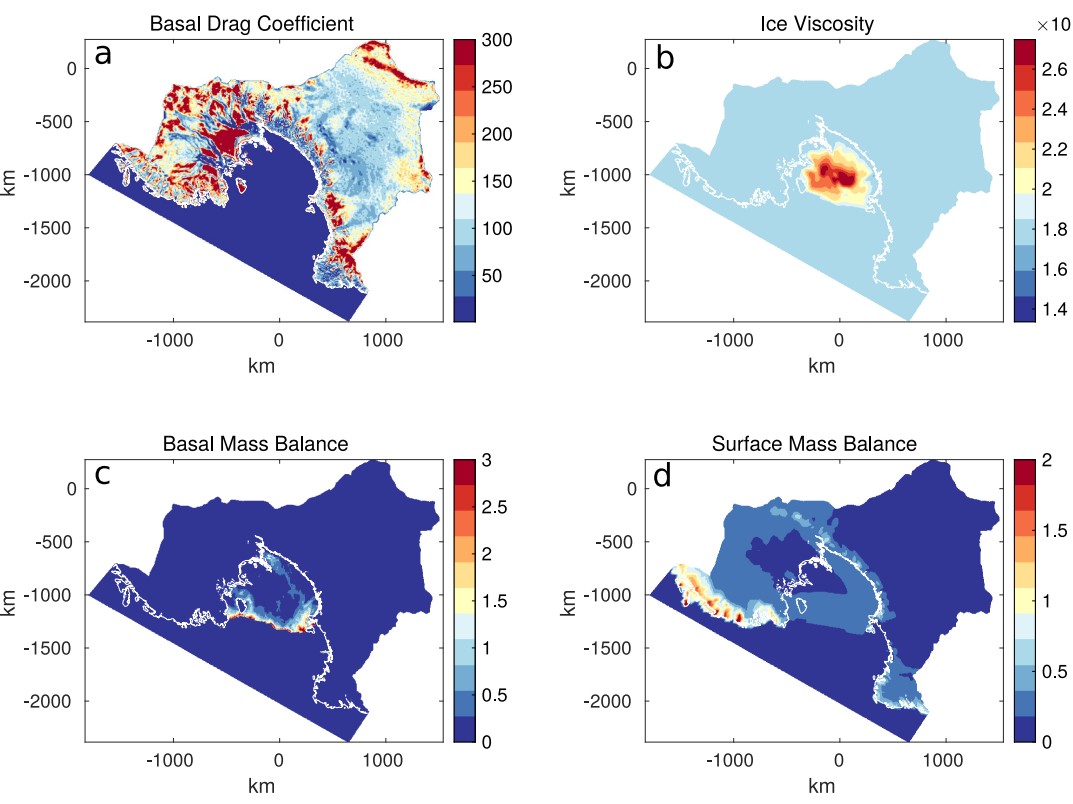

**Figure A1.** Inversions and background fields used to force the model: a) inverted basal drag coefficient ($\sqrt{(s/m)}$), b) inverted ice viscosity ($Pa\ s^{(1/n)}$), c) basal mass balance ($m/a$) and d) surface mass balance ($m/a$).

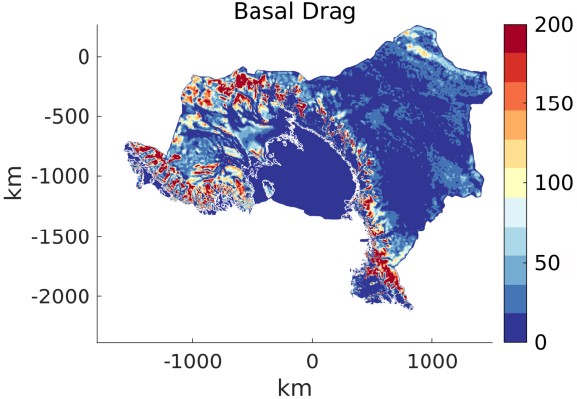

**Figure A2.** Inverted basal drag ($kPa$).

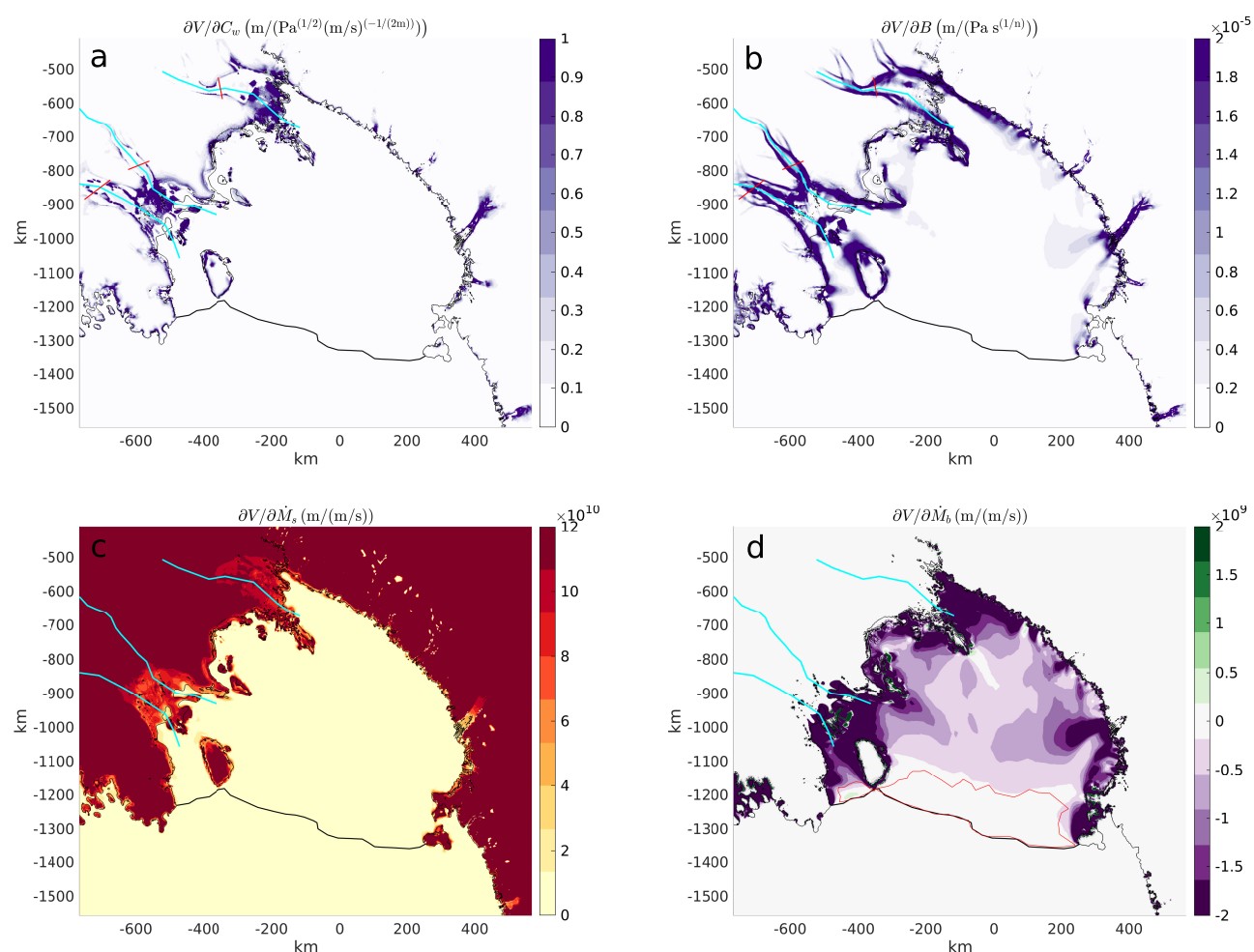

**Figure A3.** Sensitivity maps of final volume above flotation to the basal friction coefficient $C_w$ (a) and ice rigidity $B$ (b) over 40 years using the Weertman sliding law are shown in the top row of the figure. The blue lines highlight the tracks for the along-flow profiles and the red lines the across-flow profiles. Sensitivity maps of final volume above flotation to the surface mass balance $\dot{M}_s$ (c) and basal melting $\dot{M}_b$ (d) over 40 years using the Weertman sliding law are shown in the bottom row of the figure.

*Author contributions.* ISSM simulations and AD analysis were carried out by Francesca Baldacchino under the guidance of Mathieu Morlighem. AD simulations were carried out by Mathieu Morlighem. Manuscript was written by Francesca Baldacchino with contributions from Mathieu Morlighem, Nicholas Golledge, Huw Horgan and Alena Malyarenko. MITgcm basal melt outputs were provided by Alena Malyarenko.

*Competing interests.* The authors declare that they have no conflict of interest.

*Acknowledgements.* This work was supported by the New Zealand Ministry for Business, Innovation and Employment contracts RTUV1705 ("NZSeaRise") and ANTA1801 ("Antarctic Science Platform"), and Royal Society of New Zealand contract VUW-1501.

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
