# Peer review of "Sensitivity of the Ross Ice Shelf to environmental and glaciological controls"

_The Cryosphere, 2022_

## Referee Comment (RC1)

**Review of "Sensitivity of the Ross Ice Shelf to environmental and glaciological controls" by Baldacchino et al.**

**General comments**

The region of Ross Ice Shelf (Antarctica) is presently stable but basal melt rates might increase more and more in the future, leading to potential important mass loss of the ice shelf, and subsequent acceleration of its tributary glaciers. In this paper, the authors propose to map the regions of high sensitivity of the RIS basin to potential changes in ice rigidity and basal friction (glaciological controls), as well as surface mass balance and basal melting (environmental controls). For that, they use an Automatic Differentiation (AD) method, coupled to a 20-year simulation of RIS evolution under current forcings, using the Shallow-Shelf Approximation of the Ice-sheet and Sea-level System Model (ISSM). The AD allows to compute the gradient of the Volume of ice Above Floatation (VAF) with respect to the four parameters, and therefore the effect of a small perturbation in a parameter on the VAF (which is a good proxy to quantify the final effect of ice mass change on sea level rise). They conclude that the sensitivity to friction and ice rigidity is the higher at the grounding line and at glaciers and ice stream shear margins (with geographical variations). They also find, similarly to other studies (using different methods), that the sensitivity to basal melt changes is maximal at the grounding line.

The paper is relatively similar to Morlighem et al. (2021): Mapping the Sensitivity of the Amundsen Sea Embayment to Changes in External Forcings Using Automatic Differentiation, 2021). The authors used the same methods (2 co-authors are also on the 2021 paper) but on another region. As expected, the conclusions of the study are similar: the sensitivity to basal friction and ice rigidity is the stronger over the shear margins and upstream the grounding line. In this regard, little new insights are brought. However, in my opinion, the AD used by the authors is a powerful method that is still underused in the ice sheet community. Overall, this is an interesting work which can help in deciding what physical mechanism we should work on for better projections of the evolution of the region. It could also help in targeting regions to monitor when collecting observations.

Reading the results, I was wondering how different the sensitivity maps would be if (1) the forcings were different, (2) if the time period was longer (100 years for example), and (3) if the inverted friction/ice rigidity were different. If the results are sensitive to the simulation parameters and time, then, it should be discussed (see my specific comments below). If not, I think it would be worth mentioning. This could also be a great additional value with respect to Morlighem et al. (2021). Even without looking at longer simulations, I think that there was not enough results and discussion on the inversion and the 20-year simulation itself. Adding some details about it (as a supplementary material or as an Appendix) would be very valuable.

I found the comparison of the results with other papers (using totally other methods or based on observations) really interesting. I however want to point out that the discussion would benefit from being reorganized and a better writing. In general, I was slightly disappointed by the grammar and punctuation. Some sentences are poorly written and tend to decrease the readability of the paper (even in the abstract, see my technical comments).

Regardless of my concerns, and even though the method and the conclusions of the paper are very similar to Morlighem et al. (2021), I think that the relative novelty of the method and the appropriate comparison/discussion of the results with respect to other studies make the paper interesting and worthy of publication (after revision) and will be useful to the community.

**Specific comments**

- The Automatic Differentiation method used is a powerful tool but I would be careful concerning the conclusions about the sensitivity of the model to the ice rigidity change over the ice shelf. Non-linear effect aside, we could expect that a change in rigidity (let's say halving B) in a given area would lead to an important velocity change (about doubling the velocity). If such flow speed on the ice shelf was lasting longer than 20 years, it would eventually lead to a strong increase in flow from grounded ice (as the buttressing effect of the ice shelf would decrease as the ice shelf thins) and potentially subsequent VAF change. I agree that the authors clearly state that the map is the resulting sensitivity for a 20-year simulation but I think that, if you agree with my comment, it would be great to emphasis this in discussion/conclusions (similarly to what you have said for upstream ice at line 125).
- I guess that the method is also sensitive to the values of the initial parameters. The problem you solve is ill-posed by nature, which is a common problem in glaciology. During the inversion process, an underestimation of the friction can be compensated by a higher ice rigidity. How does it affect the sensitivity analysis?
- Figure 6: What is the cause of the high spikes downstream the grounding line? For SMB, this is a bit counter-intuitive given the conclusion of the sensitivity of the model to SMB changes on the ice shelf.
- Discussion:
  - While containing interesting ideas and developments (especially the numerous relations with other studies), the discussion would benefit from more structure. At the moment, ideas are a bit scattered and it is difficult to follow everything and have a proper appreciation of the results. You start with a discussion on the effect of friction, then ice rigidity before switching to SMB and basal melt rate, to come back to basal friction and rigidity. Please consider reorganizing the discussion.
  - Lines 155-165: Could you explain what is the difference between the grounding zones of the Bindschadler and MacAyeal Ice Streams and the grounding line of the Whillans Ice Stream? A few lines after, you talk about grounding line for the three ice streams.
  - Line 161: *"This stiffer till at the grounding line thickens the ice and stabilizes the position of the grounding line"*. This statement is a bit misleading. If I understand correctly the mechanism, please consider adding something like *"the stiffer till creates more basal drag and therefore a slowdown of the ice, leading to an ice thickening"* or something similar.
- The conclusion is an example of an extensive used of conjunction adverbs: however, finally, therefore, thus. I think that in general, you could delete some of these conjunctions. It could also be broken down in two paragraphs.

- Similarly, "Additionally", "in addition" or "therefore" is sometimes used twice in the same paragraph.

**Technical comments**

- Line 9-11: I found this sentence very long and maybe containing unnecessary details for an abstract. It also seems to me that it lacks a verb (?). Could you reformulate it? Maybe consider only mentioning "pinning points, larger islands, and the shear margins" instead of naming all the places.
- Line 23: consider changing *"[…] has been through ocean-forced basal melting"* to *"[…] has occurred through ocean-forced basal melting"*
- Line 30: I understand you statement about the sea level rise potential but I think that this sentence is not very clear, consider: *"These catchments, almost entirely buttressed by RIS, represent a total potential sea level rise contribution of 11.6 m"*.
- Line 44: maybe change *"at Ross Island"* for *"close to Ross Island"* (Ross Island being the grounded ice).
- Line 49: In Stewart et al. (2019), this statement is more of an outlook, as their work focuses on in-situ observations close to Ross Island. I might be wrong here but I don't think that Schodlok et al. (2016) made projections, as they focus on the comparison between observations and simulations (past and present). You might want to cite papers about projections of future basal melt rates.
- Line 57: I think you should use a comparative (larger) and not a superlative (largest)
- Line 66: delete *"here"*
- Line: 68: Could you precise what temperature field you have used? Maybe also add which sub-element scheme you used (as you mention it).
- Line 85: I am not sure to understand here. I think you mean that a 20-year forward simulation (forced by SMB and basal melt rates outputs) **IS** used in the AD package (instead of **ARE**, which would refer to the forcings).
- Line 89: Please be more specific than *"overall ice viscosity"*.
- Figure 2: It might be interesting to use a logscale colorbar here. Right know it is relatively hard to make the distinction between the higher values (dark blue) and lower (but non-null) values (light blue). Could you also use non-italic font for the units?
- Line 106: change *"[…] sensitivity of the model with respect to […]"* for *"[…] sensitivity of the model to the basal friction […]"*
- Line 107: delete *"vast"*
- Line 113: *"Increases"* instead of *"increase"*
- Line 119: add a comma between *"[…] ice rigidity"* and *"highlighting […]"*
- Line 125: add a comma between *"[…] over 20 years"* and *"therefore […]"*
- Line 132: What do you mean by *"grounding zones of the Siple Coast Ice Streams"*? To me the maximum sensitivity is much more inland, as you mentioned it before. Are you talking about the grounding zone as the surrounding of the grounding line? In this case, it does not look like the region with the highest sensitivity (maybe change the color bar so that it is visible in your Figure).
- Figure 5(a) is largely saturated, could you use a different color scale. Maybe with more than one color?

- Line 135: This statement is a bit misleading. The pinning points are very sensitive to SMB changes because they are part of the VAF. You already mentioned this for the grounded ice. You might delete this statement or put it after the statement *"This is expected since floating ice does not contribute directly to VAF"*.
- Line 149: Delete the parenthesis *"(i.e., red outline in 5 showing the 'passive' ice region)"*. Passive ice has not been introduced yet in the text.
- Line 163: To me, it is the softening of the till that leads to a basal friction change (as perceived by the model) and not the opposite. You might consider rephrasing this sentence the other way.
- Line 165: consider a new paragraph here, as you transition from a focus on the basal drag to a focus on the ice rigidity.
- Line 165: similarly, here I think you should talk about "basal friction" instead of "till conditions".
- Line 168: *"Therefore, changes in ice rigidity at the grounding zones of the Siple Coast Ice Streams  affect the ice flow and the discharge rates, which impact the overall mass balance of the RIS domain, as shown by our results."* This sentence clearly lacks punctuation (see my attempt of correction).
- Line 182: I think that your results show the opposite. In Figure 4, Bindschadler Ice Stream shows a high sensitivity to basal friction in the center of the stream. The high sensitivity at the margins is observed for the ice rigidity, or is there a mistake in the Figure labelling?
- Line 193: This is where, I think, you need another color scale on Figure 5.
- Line 197: It is only now that I really understand your statement at line 132. I think you need to be clearer at line 132 when talking about the high sensitivity of the grounding zone.
- Line 200: I am not sure that "promote ice thickness" is really what you want to say, maybe only "modify" or "affect". For example: "[…] would affect ice thickness and discharge on the RIS"
- Line 206: Maybe I am misunderstanding but the effect of the ocean heat on the ice temperature and the rate of basal melting is not really accounted for in the model, right? You apply the melt as a forcing with no feedback. It would be nice to specify this here.
- Line 212: would have greatest impact (compared to what?) on the final VAF if what? If basal melting was increasing in the future?
- Line 265: You write: *"These results show that the RIS mass balance is highly sensitive to changes in ocean circulation and mixing as well as tidal currents which could potentially drive changes in basal melt rates at the calving front at the grounding zones."* I guess you mean "at the calving front **AND** ant the grounding zones"?
- Line 268: I would delete the end of the sentence *"as this study identifies that these regions are important for the mass balance of the RIS"* as you already explained these two sentences earlier.
- Line 269: I agree with you. Such modeling can be useful to plan field campaigns. We do not use enough this kind of tool.

---

## Referee Comment (RC2)

**Review of, "Sensitivity of the Ross Ice Shelf to environmental and glaciological controls", by Baldacchino et al.**

**Summary**

In this article, the authors report on a sensitivity study of the short-term (20 years) mass balance of the ice sheet catchments surrounding the Ross Ice Shelf to perturbations in surface and basal mass balance, basal drag, and ice rheology.  They use the automatic differentiation capability within the Ice Sheet System Model to evaluate the linear sensitivity of the 20-year Volume Above Flotation (VAF) to each of those four spatially distributed parameters- that is, the derivative of VAF (evaluated after 20 years) with respect to localized perturbations in those four spatially distributed parameters.  With respect to the two force-balance variables they tested, they found high sensitivity of VAF to basal drag within the fast-flowing but low-drag downstream regions of the Siple Coast ice streams, as well as high sensitivity within the main trunk of Byrd Glacier, while the sensitivity of VAF to ice rigidity was highest in shear margins and around the perimeter Roosevelt Island.  With respect to the two mass-balance variables they tested, they found near uniform sensitivity of VAF to SMB within the majority of the grounded ice sheet, no doubt reflecting the short 20-year timepan of their simulation, while the sensitivity to BMB was greatest near the grounding line.

The method used by the authors is sensible, the results are well presented and well discussed, and the conclusions are supported by the results presented.  This paper describes the application of an existing technique to a new area of the ice sheet, and thus represents a solid incremental advance in our understanding of the Antarctic Ice Sheet.  This paper deserves to be published in *The Cryosphere*.  However, before publication, I have some concerns that I would like to see addressed around the analysis of basal drag.  I describe these concerns next, and then move on to more minor comments.  Overall, my concerns can be addressed with additional figures (or additional subplots in the existing figures) and a bit more analysis.

**Major Comment: Basal Drag vs Basal Drag Coefficient**

My biggest concern with this paper is with the use of a linear sliding law, in which basal drag is directly proportional to sliding velocity.  Simply put, a linear sliding law is not a realistic or physically defensible representation of dynamics at the ice base.  Even in areas of the ice sheet underlain by hard bedrock, the sliding relationship is expected to be multi-valued with a maximum basal drag given by Iken's bound (Iken, 1981; Schoof, 2005; Gagliardini et al., 2007), a relationship which is quite different from a linear one (especially for fast sliding, like the ice streams where the authors found high sensitivity).  Moreover, the Siple Coast ice streams, where the authors find the highest sensitivity to basal drag, are known to be underlain not by hard bedrock but by soft subglacial till that obeys a Coulomb plastic friction law (Tulaczyk et al., 2000).  The linear sliding rule used by the authors corresponds to power-law sliding with an exponent equal to one, while a more realistic Coulomb plastic sliding law corresponds to the limit where the exponent approaches infinity.  Thus, the sliding law used in this paper is the exact opposite of a realistic one for the region of interest.  Therefore, it is critical that the authors present results that are robust to the choice of sliding law.

My recommendation is that the authors show maps of the sensitivity of VAF to basal drag ($\tau_b$) alongside their maps of sensitivity to drag coefficient.  The drag coefficient ($C_b$) is an artifact of the choice of a linear sliding law, while in a more realistic Coulomb plastic sliding law, the basal yield stress would be the quantity which describes the state of the bed, and the basal yield stress would be equal to the basal drag in areas that are actively sliding.  Furthermore, while the coefficient ($C_b$) will change with different sliding laws, the basal drag ($\tau_b$) is likely to be broadly similar for all sliding laws for two reasons: 1) all potential inversions with all possible sliding laws would be run within the same ice sheet geometry, and thus experience the same gravitational driving stress; and, 2) all potential inversions with all possible sliding laws would be constrained to

match the same observed surface velocities, and thus all inversions will converge on similar patterns of englacial stress transmission.  When identical patterns of driving stress are combined with similar patterns of englacial stress transmission, the result is that momentum conservation will dictate similar basal drag patterns regardless of the sliding law.

Fortunately, it should be possible to compute the sensitivity of VAF to τ from their existing results on the sensitivity to C, without the need to recompute the entire model. Starting with the linear sliding relationship (Eq. 2 in the manuscript), we have,

$$\tau_b = C_b^2 N v_b \quad, \tag{1}$$

where N is effective pressure and $v_b$ is the basal sliding velocity.  We can then compute the derivative of drag with respect to drag coefficient,

$$\frac{d\tau_b}{dC_b} = 2 C_b N v_b \quad. \tag{2}$$

Once we have the derivative of drag with respect to drag coefficient, it is easy to convert the sensitivity estimate the authors already have (dV/dC) to one that would be more robust to the choice of sliding law (dV/dτ), like so,

$$\frac{dV}{d\tau_b} = \frac{dV}{dC_b} \frac{dC_b}{d\tau_b} = \frac{1}{2 C_b N v_b} \frac{dV}{dC_b} \quad. \tag{3}$$

This quantity, the sensitivity of ice sheet mass balance to changes in basal drag, is likely to be far more representative of the soft sediments that compose the true ice sheet bed than the sensitivity to drag coefficient.  A map showing this quantity should be added to figure 2, and this quantity should also replace the sensitivity to drag coefficient in the top rows of figures 3 and 4. Conclusions drawn based on the sensitivity to basal drag are more likely to be robust to changes in the sliding law than conclusions drawn based on the sensitivity to drag coefficient.

In addition, it might also be interesting to look at maps of the sensitivity to normalized perturbations in basal drag, beyond merely the sensitivity to drag perturbations.  That is to say, it might be interesting to look at the sensitivity to dτ/τ, rather than merely dτ.  That is because there is a very large range in basal drag within the domain: within the very low-friction ice plains in the downstream regions of the Siple Coast ice streams, an increase in τ by a few kPa could easily represent a doubling of the local basal drag, whereas in other regions that same perturbation might be an increase of only a few percent.  Using Equations 1 and 3, the sensitivity to a relative drag perturbation can be computed by,

$$\frac{dV}{d\tau_b / \tau_b} = \tau_b \frac{dV}{d\tau_b} = \frac{C_b}{2} \frac{dV}{dC_b} \quad. \tag{4}$$

However, I view the sensitivity to relative drag perturbations (4) to be less important than the basic sensitivity to drag perturbations (3).  If the authors want to omit the sensitivity to relative drag perturbations, that is fine.  It is only important that the sensitivity analysis by done with respect to drag, rather than drag coefficient, to ensure that the results of this study are robust to the choice of sliding law.  Of course, I do not expect the results to be hugely different.  It is likely that the map of sensitivity to drag will be similar in spatial structure to the map of sensitivity to drag coefficient, so the authors probably will not need to change their conclusions or rewrite much of the manuscript. However, until the map is actually made, we cannot know for sure.  An analysis of the sensitivity of the model to basal drag will be much more physically defensible and more robust to changes in sliding law than an analysis of the sensitivity of the model to the coefficient of a linear sliding law.

**Other Comments**

L29-30: "Through the ice shelf restraining this ice in the catchment, it has a total potential contribution to sea level rise of 11.6 m (Tinto et al., 2019)."
    This is very awkwardly worded. Perhaps rephrase to something like, "The grounded ice in its catchment has the potential to raise sea level by up to 11.6 m (Tinto et al., 2019)."

Figure 1
    Perhaps it would be better to show velocity with a logarithmic color scale? Velocity varies by several orders of magnitude over the ice sheet. Presenting it on a linear scale as in this figure has the effect of emphasizing fast-flowing areas, especially the ice shelf. Perhaps this was the intention; however, a side effect of this is that the structure of ice streams in the grounded part of the domain is faded and difficult to see in many places. In addition, it would be helpful to show some aspects of the ice sheet geometry in additional plots of Figure 1, such as the bed elevation, ice thickness, or surface slope.

L68: "...ice viscosity depending on the ice temperature…"
    Where do you get the ice temperature from?

L72-83: Model setup
    What does your inverted drag coefficient look like? What about the other forcing fields? It would be useful to have a figure showing maps of the background fields used to force the model, so that we can put the sensitivity maps in context. I would be interested in seeing a figure that showed the inverted basal drag coefficient, the actual basal drag, the ice rigidity, surface mass balance, and basal mass balance. Without being able to see these fields, it is hard to put the sensitivity maps in context. Remember, the quantity being computed by the AD procedure is the linear sensitivity- that is, the derivative of the output quantity with respect to small perturbations in the input quantity, *evaluated at the given value of the input quantity*. If the background ice sheet configuration changes, the sensitivity will change as well.

L102: Equation 4, compared to Figures 2 and 5
    This equation implies that you have the units wrong in the labels of Figures 2 and 5. In order to get from the sensitivity to a parameter, $DV(P)$ and the perturbation in that parameter, $\delta P$, to an estimate in the volume change (V), it is necessary to perform a spatial integral over the model domain, with differential $d\Omega$. Thus, the sensitivities $DV(P)$ should have units of meters per (parameter units), rather than $m^3$ per (parameter units), as you have given in the figure titles. The extra $m^2$ necessary to form a volume comes from the $d\Omega$ in the integral.

L107: "These sensitivity maps show that the vast majority of the grounded ice is not sensitive to changes in friction or rheology…"
    This wording isn't quite right. Presumably, the local flow in the vast majority of the grounded ice is going to be sensitive to local changes in friction or rheology. What you mean here is what you explained in the rest of the sentence, namely, that the overall mass balance of the ice sheet (after 20 years) is not sensitive to local changes in friction or rheology in the vast majority of the grounded domain. Perhaps a better wording would be, "These maps show that the sensitivity to friction and rheology is low in the vast majority of the grounded domain…"

L118-120: "Kamb Ice Stream on the Siple Coast shows low to no sensitivity to changes in the basal friction or ice rigidity highlighting that the Kamb Ice Stream is currently stagnant and thus changing the friction or ice rigidity will not change the ice discharge significantly."
    It is worth pointing out that this result is a byproduct of the assumptions inherent in a linear perturbation analysis, and thus this negative result for Kamb ice stream can potentially be

misleading. The sensitivity maps produced by this method basically answer the question: 'if we changed the friction coefficient at this particular location by an infinitessimal amount, what would be the marginal change in VAF?' In the continuous limit, each perturbation would only be to a single point with vanishing area, while in the numerical implementation, the perturbation resides at a single mesh node.

Thus, the method cannot capture the sensitivity of VAF to a general reactivation of Kamb ice stream, which would involve a spatially correlated large-amplitude reduction in basal friction across the entire bed of the former ice stream. Clearly, a general reactivation of Kamb ice stream would reduce the grounded volume of the ice sheet. However, a reduction in basal friction at one location within the former ice stream would not have much effect if friction remained high across the rest of Kamb. Thus, the result of a low sensitivity within Kamb Ice Stream is accurate within the assumptions of this method, but it can be misleading in the sense that this method doesn't test for the possibility of spatially correlated perturbations to model parameters, and a general ice stream reactivation would be a spatially correlated perturbation.

As I mentioned above, this method tests the sensitivity of VAF to perturbations in the forcing parameters, *assuming that the general ice sheet configuration does not change*. The reactivation of a dormant ice stream is a big enough perturbation that it would constitute a change in the general ice sheet configuration.

Figure 3:
Are the negative values in the bottom row (sensitivity to rheology B) real, or do these simply represent numerical artifacts? Put another way, do you actually believe that there are places where strengthening the ice will actually cause *more* of it to flow into the ocean? There are also some negative values in the top row of this figure and in figure 4, but those negative excursions are much smaller than the negative excursions in the bottom row of figure 3.

Figure 5, L127-149, L192-201: Sensitivity to SMB and discussion of sensitivity to SMB
I would be very interested in seeing a map of the sensitivity to SMB normalized by the stagnant ice sensitivity- that is, the sensitivity that you would get assuming that the ice did not move and the perturbation to SMB simply piled up mass at the location of the perturbation. That "stagnant ice" sensitivity is simply given by the time period of the simulation, in this case 20 years (by the way, I believe that you have made an additional units error in Figure 5, beyond the units error I pointed out earlier: the sensitivities to surface and basal mass balance probably should have units of m/(m/s), rather than m/(m/a) as you put in Figure 5, since otherwise values on the order of $6-7 \times 10^8$ are far too large. However, if ISSM is producing sensitivities in terms of m/s instead of m/a, then those magnitudes are exactly what one would expect from the stagnant-ice sensitivity with a 20 year model runtime). Normalizing the SMB sensitivity by the stagnant-ice sensitivity should help to put the SMB sensitivity in context. If there are any areas with sensitivity greater than the stagnant-ice value, then you could say definitively that SMB is vital to keeping those areas grounded and buttressing the rest of the ice sheet. Areas below the stagnant-ice value probably reflect regions where ice flux evolves very quickly (within the 20 year runtime) to export additional mass.

I also would expect the results for SMB sensitivity to drop below the stagnant-ice sensitivity as model run time is increased beyond 20 years. In a steady state, the sensitivity of ice volume to SMB is quite weak, as increases in SMB are mostly balanced by increases in flow rate into the ocean, with only small increases in surface slope (and thus inland ice thickness) necessary to produce those increased flow rates. A classic scaling analysis from John Nye suggests that the steady-state thickness of an ice sheet scales with accumulation rate to the power of $1/(2m+1)$, where $m$ is the sliding exponent (Nye, 1959). Thus, even with a linear sliding law ($m=1$), thickness would only increase with the cube root of accumulation rate, while for more realistic sliding laws the dependence would be weaker still, and for a Coulomb plastic bed where $m \to \infty$, such as the bed found in the Siple Coast ice streams, the steady-state sensitivity should approach zero. But with

only a 20 year model runtime, I would expect the vast majority of the inland ice to have a constant SMB sensitivity given by the stagnant-ice value.

L157-158: "Our results show that the Bindschadler and MacAyeal Ice Streams have high sensitivities to changes in basal friction at their grounding zones due to these 'sticky spots' being key in controlling the ice discharge"

Do your results in fact show high sensitivity at the sticky spots, as opposed to high sensitivity in the ice plains generally? This is where it would be helpful to have a map of the inversion results to compare your sensitivity maps to. In addition, this is where it would be helpful to distinguish between the sensitivity to perturbations in drag and the sensitivity to relative perturbations in drag, as I discussed earlier.

**References:**

Gagliardini, O., Cohen, D., Råback, P., and Zwinger, T.: Finite-element modeling of subglacial cavities and related friction law, J. Geophys. Res. Earth Surf., 112, https://doi.org/10.1029/2006JF000576, 2007.

Iken, A.: The effect of the subglacial water pressure on the sliding velocity of a glacier in an idealized numerical model, J. Glaciol., 27, 407–421, 1981.

Nye, J. F.: The motion of ice sheets and glaciers, J. Glaciol., 3, 493–507, 1959.

Schoof, C.: The effect of cavitation on glacier sliding, Proc. R. Soc. Lond. Math. Phys. Eng. Sci., 461, 609–627, https://doi.org/10.1098/rspa.2004.1350, 2005.

Tulaczyk, S., Kamb, W. B., and Engelhardt, H. F.: Basal mechanics of Ice Stream B, West Antarctica: 1. Till mechanics, J. Geophys. Res. Solid Earth, 105, 463–481, https://doi.org/10.1029/1999JB900329, 2000.

---

## Author Comment (AC1)

**Sensitivity of the Ross Ice Shelf to environmental and glaciological controls -Response to reviewers-**

Francesca BALDACCHINO et al

June 08 2022

We would like to thank the two anonymous reviewers and the editor for their positive and constructive comments. We address their remarks below point by point.

**1 Reviewer #1**

Regardless of my concerns, and even though the method and the conclusions of the paper are very similar to Morlighem et al. (2021), I think that the relative novelty of the method and the appropriate comparison/discussion of the results with respect to other studies make the paper interesting and worthy of publication (after revision) and will be useful to the community.

We would like to thank reviewer for their positive review and excellent suggestions. It is true that this work builds on Morlighem et al. (2021). We use a similar methodology here, but instead of focusing on the Amundsen Sea Embayment that has been changing rapidly, we are looking at a large cold ice shelf.

**1.1 Specific Comments**

- The Automatic Differentiation method used is a powerful tool but I would be careful concerning the conclusions about the sensitivity of the model to the ice rigidity change over the ice shelf. Non-linear effect aside, we could expect that a change in rigidity (let's say halving B) in a given area would lead to an important velocity change (about doubling the velocity). If such flow speed on the ice shelf was lasting longer than 20 years, it would eventually lead to a strong increase in flow from grounded ice (as the buttressing effect of the ice shelf would decrease as the ice shelf thins) and potentially subsequent VAF change. I agree that the authors clearly state that the map is the resulting sensitivity for a 20-year simulation but I think that, if you agree with my comment, it would be great to emphasis this in discussion/conclusions (similarly to what you have said for upstream ice at line 125). We ran the model again with a longer time period (40 years instead of 20 years) and with a Weertman friction law to investigate the sensitivity of the results to these choices. The results can be found Figures 1 and 2 in this document. We found that the sensitivity of our results to the length of the transient simulation and friction law are qualitatively similar, suggesting that the areas highlighted here are strong features. Extending the length of the transient simulation multiplies the gradient of sensitivities accordingly (i.e. doubling the simulation time leads to a doubling of the sensitivity) but the sensitivity patterns remain quantitatively similar to a shorter transient simulation. This was also found in Morlighem et al., 2021 whom stated "The consistency of these sensitivities across different models demonstrates that our conclusions are related to the physics of the system, and not due to specific modeling approaches or model parameters." These results will be included in the appendix of the revised manuscript. The 40 year simulation results for the Budd linear sliding law will be used within the final manuscript results section and discussion. In addition, an additional paragraph will be included at the end of the discussion: " Our model relies on a Budd linear sliding law, in which basal drag is directly proportional to sliding velocity. This friction law may not be valid under some sectors of our model domain such as the Siple Coast. We performed additional experiments to test the sensitivity of our results to the friction law and length of transient simulation (Figures A3 and A4 in the Appendix) by using a Weertman friction law instead. We found similar conclusions to Morlighem et al., 2021 with the Weertman friction law showing quantitatively similar results with the Budd sliding law suggesting that the areas highlighted in the sensitivity maps are robust features. Extending the length of the transient simulation increases the sensitivities accordingly (i.e. doubling the simulation time will double the sensitivity) but the sensitivity patterns remain quantitatively similar to the 20 year simulation. These results suggest that the conclusions drawn from the sensitivity maps are reliable and related to the physics of the system not the model parameters Morlighem et al., 2021." In addition, the length of the simulation (i.e. 40 years) will be emphasised in the results, discussion and conclusion as suggested by reviewer.

- I guess that the method is also sensitive to the values of the initial parameters. The problem you solve is ill-posed by nature, which is a common problem in glaciology. During the inversion process, an underestimation of the friction can be compensated by a higher ice rigidity. How does it affect the sensitivity analysis?

The inversion results for ice rigidity and basal friction will be included in the appendix of the final manuscript (Figure 3 in this document). Morlighem et al., 2021 showed that the sensitivity maps are not significantly dependent on the initial parameters: ISSM and STREAMICE produced the the same sensitivities (in Amundsen Sea Embayment) even though each model used different stress balance equations, meshes, discretization methods and initialization procedures. As stated in Morlighem et al., 2021: "The agreement

between the sensitivity maps produced by the two ice-sheet models that are completely independent indicates the robustness of the results."

- Figure 6: What is the cause of the high spikes downstream the grounding line? For SMB, this is a bit counter-intuitive given the conclusion of the sensitivity of the model to SMB changes on the ice shelf.

The high spikes found in Figure 6 are most likely numerical artefacts due to our numerical model's treatment of the grounding line following a sub-element grid scheme (i.e. subelement paramterization). Morlighem et al 2021 also observed that ISSM shows a strong sensitivity to melt at the grounding line, when theory shows that the sensitivity should be zero. They suggested that any sensitivity within 1-element of the grounding line should be interpreted with caution. This will be added to our results: "The high spikes observed in Figure 6 downstream of the grounding lines are numerical artefacts and should be interpreted with caution [Morlighem et al., 2021]. These numerical artefacts are likely due to the sub-element parameterization used within the numerical model's treatment of the grounding line."

- While containing interesting ideas and developments (especially the numerous relations with other studies), the discussion would benefit from more structure. At the moment, ideas are a bit scattered and it is difficult to follow everything and have a proper appreciation of the results. You start with a discussion on the effect of friction, then ice rigidity before switching to SMB and basal melt rate, to come back to basal friction and rigidity. Please consider reorganizing the discussion.

The discussion will be restructured and hopefully the reviewer agrees that the ideas are less scattered. The discussion will be structured (numbers relate to new paragraphs) like this: 1) Basal friction: Siple Coast Ice Streams, 2) Ice Rigidity: Siple Coast Ice Streams, 3) Basal friction and Ice rigidity: Byrd Glacier, 4) Basal friction and Ice rigidity: Pinning points, 5) SMB for model domain, 6) Basal melt: Siple Coast Ice Streams, 7) Basal melt: Byrd Glacier, 8) Basal melt: Pinning points, 9) Basal melt: Calving front.

- Lines 155-165: Could you explain what is the difference between the grounding zones of the Bindschadler and MacAyeal Ice Streams and the grounding line of the Whillans Ice Stream? A few lines after, you talk about grounding line for the three ice streams.

This is a good point and we misused the term "grounding zone". This will be clarified and reworded in Lines 111-112: "in the vicinity of the grounding line". In addition, throughout the manuscript grounding zones will be removed, clarified and reworded using the above sentence and "grounding line".

- Line 161: "This stiffer till at the grounding line thickens the ice and stabilizes the position of the grounding line". This statement is a bit misleading. If I understand correctly the mechanism, please consider adding something like "the stiffer till creates more basal drag and therefore a slowdown of the ice, leading to an ice thickening" or something similar. This sentence will be reworded accordingly so the statement is no longer misleading. "The stiffer till at the grounding line creates more basal drag and consequently a slowdown of ice, leading to an thickening of ice which stabilizes the position of the grounding line."

- The conclusion is an example of an extensive used of conjunction adverbs: however, finally, therefore, thus. I think that in general, you could delete some of these conjunctions. It could also be broken down in two paragraphs.

Finally and therefore will be deleted as well as the conclusion being split into two paragraphs.

- Similarly, "Additionally", "in addition" or "therefore" is sometimes used twice in the same paragraph.

The repetitive use of "Additionally", "in addition" or "therefore" will be removed or replaced throughout the manuscript.

**1.2** Technical Comments**

- Line 9-11: I found this sentence very long and maybe containing unnecessary details for an abstract. It also seems to me that it lacks a verb (?). Could you reformulate it? Maybe consider only mentioning "pinning points, larger islands, and the shear margins" instead of naming all the places.

The sentence will be reworded: "With changes in basal melting close to the grounding lines of the Siple Coast Ice Streams and Transantarctic Mountains Outlet Glaciers having a larger impact on the final VAF compared to elsewhere. Additionally, the pinning points and ice shelf shear margins are highly sensitive to changes in basal melt."

- Line 23: consider changing "[...] has been through ocean-forced basal melting" to "[...] has occurred through ocean-forced basal melting"

This will be done.

- Line 30: I understand your statement about the sea level rise potential but I think that this sentence is not very clear, consider: "These catchments, almost entirely buttressed by RIS, represent a total potential sea level rise contribution of 11.6 m".

This will be done.

- Line 44: maybe change "at Ross Island" for "close to Ross Island" (Ross Island being the grounded ice).

This will be done.

- Line 49: In Stewart et al. (2019), this statement is more of an outlook, as their work focuses on in-situ observations close to Ross Island. I might be wrong here but I don't think that Schodlok et al. (2016) made projections, as they focus on the comparison between observations and simulations (past and present). You might want to cite papers about projections of future basal melt rates.

Appropriate references will be added and sentence reworded: "Summer sea-ice concentrations in the Ross Sea are projected to decreases by 56% by 2050 [Smith Jr. et al., 2014] with this ice-free period also expected to increase [Dinniman et al., 2018] which will highly likely increase ice-shelf basal melting impacting the future stability of the RIS [Stewart et al., 2019]."

- Line 57: I think you should use a comparative (larger) and not a superlative (largest).

This will be done.

- Line 66: delete "here".

This will be done.

- Line: 68: Could you precise what temperature field you have used? Maybe also add which sub-element scheme you used (as you mention it).

These sentences will be added to methods: "We use the ISSM Ice Sheet Model Intercomparison Project for CMIP6 (ISMIP6) temperature field to initialize the ice viscosity over floating ice." In addition, the sub-element scheme used will be added (i.e. Sub element Parameterization 1).

- Line 85: I am not sure to understand here. I think you mean that a 20-year forward simulation (forced by SMB and basal melt rates outputs) IS used in the AD package (instead of ARE, which would refer to the forcings).

This sentence will be clarified.

- Line 89: Please be more specific than "overall ice viscosity".

This will be done.

- Figure 2: It might be interesting to use a logscale colorbar here. Right now it is relatively hard to make the distinction between the higher values (dark blue) and lower (but non-null) values (light blue). Could you also use non-italic font for the units?

We explored using a logscale colorbar here however decided against it. The logscale bar did not make the distinction clearer and resulted in Figure 2b (i.e. Ice Rigidity) distinctions being less clear. In addition, Morlighem et al 2021 does not use a logscale bar for these results and thus enables better comparison. The font for the units will be changed to non-italics.

- Line 106: change "[...] sensitivity of the model with respect to [...]" for "[...] sensitivity of the model to the basal friction [...]"

This will be done.

- Line 107: delete "vast"

This will be done.

- Line 113: "Increases" instead of "increase"

This will be done.

- Line 119: add a comma between "[...] ice rigidity" and "highlighting [...]"

This will be done.

- Line 125: add a comma between "[...] over 20 years" and "therefore [...]"

This will be done.

- Line 132: What do you mean by "grounding zones of the Siple Coast Ice Streams"? To me the maximum sensitivity is much more inland, as you mentioned it before. Are you talking about the grounding zone as the surrounding of the grounding line? In this case, it does not look like the region with the highest sensitivity (maybe change the color bar so that it is visible in your Figure).

The term "grounding zones" will be removed and clarification will be added to Line 132 and the color bar will be changed in Figure 5(a).

- Figure 5(a) is largely saturated, could you use a different color scale. Maybe with more than one color?

A different color scale will be used on Figure 5(a). Figure 4 in this document show the changes that will be made.

- Line 135: This statement is a bit misleading. The pinning points are very sensitive to SMB changes because they are part of the VAF. You already mentioned this for the grounded ice. You might delete this statement or put it after the statement "This is expected since floating ice does not contribute directly to VAF".

This will be put after the statement: "This is expected since floating ice does not contribute directly to VAF".

- Line 149: Delete the parenthesis "(i.e., red outline in 5 showing the 'passive' ice region)". Passive ice has not been introduced yet in the text.

This will be done.

- Line 163: To me, it is the softening of the till that leads to a basal friction change (as perceived by the model) and not the opposite. You might consider rephrasing this sentence the other way.

This will be done.

- Line 165: consider a new paragraph here, as you transition from a focus on the basal drag to a focus on the ice rigidity.

Discussion will be restructured with separate paragraphs for basal drag and ice rigidity changes.

- Line 165: similarly, here I think you should talk about "basal friction" instead of "till conditions".

This will be done.

- Line 168: "Therefore, changes in ice rigidity at the grounding zones of the Siple Coast Ice Streams changes affect the ice flow and the discharge rates, which impacts the overall mass balance of the RIS domain, as shown by our results." This sentence clearly lacks punctuation (see my attempt of correction).

The wording of the sentence will be changed: "Therefore, our results show that changes in ice rigidity at the grounding lines of the Siple Coast Ice Streams affect the ice flow and discharge rates impacting the overall mass balance of the RIS domain."

- Line 182: I think that your results show the opposite. In Figure 4, Bindschadler Ice Stream shows a high sensitivity to basal friction in the center of the stream. The high sensitivity at the margins is observed for the ice rigidity, or is there a mistake in the Figure labelling?

There is no mistake in labelling of the figure. We were commenting on the small spikes in sensitivity in the Bindschadler Ice Stream margins to basal friction that we can see in Figure 4. We will reword the sentence on Line 182 from "highly sensitive" to "sensitive".

- Line 193: This is where, I think, you need another color scale on Figure 5.

This will be done.

- Line 197: It is only now that I really understand your statement at line 132. I think you need to be clearer at line 132 when talking about the high sensitivity of the grounding zone.

The phrase "grounding zones" will be removed and clarified/reworded throughout the manuscript.

- Line 200: I am not sure that "promote ice thickness" is really what you want to say, maybe only "modify" or "affect". For example: "[...] would affect ice thickness and discharge on the RIS"

This will be changed from "promote" to "affect".

- Line 206: Maybe I am misunderstanding but the effect of the ocean heat on the ice temperature and the rate of basal melting is not really accounted for in the model, right? You apply the melt as a forcing with no feedback. It would be nice to specify this here.

An extra sentence will be added at Line 206: "Basal melt is applied as a forcing with no feedback in the numerical model resulting in the sensitivity maps not taking into account the effect of ocean heat or rate of basal melting."

- Line 212: would have greatest impact (compared to what?) on the final VAF if what? If basal melting was increasing in the future?

We will reword Line 212: "This highlights that changes in basal melting are currently occurring in sensitive areas that have the greatest impact on the final VAF compared to elsewhere."

- Line 265: You write: "These results show that the RIS mass balance is highly sensitive to changes in ocean circulation and mixing as well as tidal currents which could potentially drive changes in basal melt rates at the calving front at the grounding zones." I guess you mean "at the calving front AND at the grounding zones"?

This will be changed.

- Line 268: I would delete the end of the sentence "as this study identifies that these regions are important for the mass balance of the RIS" as you already explained these two sentences earlier.

This will be done.

-Line 269: I agree with you. Such modeling can be useful to plan field campaigns. We do not use enough this kind of tool.

Exactly! Thanks for your comments.

**2 Reviewer #2**

The method used by the authors is sensible, the results are well presented and well discussed, and the conclusions are supported by the results presented. This paper describes the application of an existing technique to a new area of the ice sheet, and thus represents a solid incremental advance in our understanding of the Antarctic Ice Sheet. This paper deserves to be published in The Cryosphere. However, before publication, I have some concerns that I would like to see addressed around the analysis of basal drag. I describe these concerns next, and then move on to more minor comments. Overall, my concerns can be addressed with additional figures (or additional subplots in the existing figures) and a bit more analysis.

We would like to thank reviewer for their constructive review and excellent suggestions which have improved the manuscript.

**2.1 Major Comments**

-My biggest concern with this paper is with the use of a linear sliding law, in which basal drag is directly proportional to sliding velocity. Simply put, a linear sliding law is not a realistic or physically defensible representation of dynamics at the ice base. Even in areas of the ice sheet underlain by hard bedrock, the sliding relationship is expected to be multi-valued with a maximum basal drag given by Iken's bound (Iken, 1981; Schoof, 2005; Gagliardini et al., 2007), a relationship which is quite different from a linear one (especially for fast sliding, like the ice streams where the authors found high sensitivity). Moreover, the Siple Coast ice streams, where the authors find the highest sensitivity to basal drag, are known to be underlain not by hard bedrock but by soft subglacial till that obeys a Coulomb plastic friction law (Tulaczyk et al., 2000). The linear sliding rule used by the authors corresponds to power-law sliding with an exponent equal to one, while a more realistic Coulomb plastic sliding law corresponds to the limit where the exponent approaches infinity. Thus, the sliding law used in this paper is the exact opposite of a realistic one for the region of interest. Therefore, it is critical that the authors present results that are robust to the choice of sliding law. My recommendation is that authors show maps of the sensitivity of VAF to basal drag alongside their maps of sensitivity to drag coefficient. This quantity, the sensitivity of ice sheet mass balance to changes in basal drag, is likely to be far more representative of the soft sediments that compose the true ice sheet bed than the sensitivity to drag coefficient. A map showing this quantity should be added to figure 2, and this quantity should also replace the sensitivity to drag coefficient in the top rows of figures 3 and 4.

Firstly, we would like to thank the reviewer for their extremely useful comments regarding the sliding law used and the time they spent on this issue. The reviewer is recommending this due to their concern regarding the use of the Budd linear sliding law. Therefore, we decided to rerun the simulations using the Weertman Friction Law. Morlighem et al 2021 also compared the Budd linear sliding law with the Weertman Friction law and found similar results for their location of interest. The sensitivity maps for the Weertman Friction law will be included in the Appendix of the final manuscript (Figures 1 and 2 in this document). We can see that the sensitivity of our results to the friction law are qualitatively similar, suggesting that the areas highlighted are robust and reliable features. The largest differences are found in the ice rigidity and basal friction spatial distributions with the Weertman sliding law displaying larger sensitivities to ice rigidity (especially along ice the ice stream shear margins) and smaller sensitives to basal friction (especially within the main ice trunks). However, these sensitives are marginally different to the Budd sliding law and display similar spatial patterns highlighting that these identified sensitive regions are strong features that need to be monitored in the future. Therefore, we continue to use the sensitivity to drag coefficient in the manuscript as analysis of the AD results as we have shown that the use of the Budd linear sliding law is robust for the model sensitivity.

- In addition, it might also be interesting to look at maps of the sensitivity to normalized perturbations in basal drag, beyond merely the sensitivity to drag perturbations. That is to say, it might be interesting to look at the sensitivity to  $d\tau/\tau$ , rather than merely  $d\tau$ . That is because there is a very large range in basal drag within the domain: within the very low-friction ice plains in the downstream regions of the Siple Coast ice streams, an increase in  $\tau$  by a few kPa could easily represent a doubling of the local basal drag, whereas in other regions that same perturbation might be an increase of only a few percent. However, I view the sensitivity to relative drag perturbations to be less important than basic sensitivity to drag perturbations. If the authors want to omit the sensitivity to relative drag perturbations, that is fine. It is only important that the sensitivity analysis by done with respect to drag, rather than drag coefficient, to ensure that the results of this study are robust to the choice of sliding law. Of course, I do not expect the results to be hugely different. It is likely that the map of sensitivity to drag will be similar in spatial structure to the map of sensitivity to drag coefficient, so the authors probably will not need to change their conclusions or rewrite much of the manuscript. However, until the map is actually made, we cannot know for sure. An analysis of the sensitivity of the model to basal drag will be much more physically defensible and more robust to changes in sliding law than an analysis of the sensitivity of the model to the coefficient of a linear sliding law.

This is an interesting idea but we ultimately decided not to plot the normalized sensitivities and to show the results of another friction law instead. We decided to do this as we feel that including the normalized sensitivities would not have added significant value to the discussion and may cause confusion for the readers regarding which figures to use for identifying areas of sensitivity and concern for future field campaigns. The normalised basal friction sensitivities can be found in Figure 4 for the Budd linear sliding law.

**2.2 Other Comments**

- L29-30: "Through the ice shelf restraining this ice in the catchment, it has a total potential contribution to sea level rise of 11.6 m (Tinto et al., 2019)." This is very awkwardly worded. Perhaps rephrase to something like, "The grounded ice in its catchment has the potential to raise sea level by up to 11.6 m (Tinto et al., 2019)."

We will rephrase to: "These catchments, almost entirely buttressed by RIS, represent a total potential sea level rise contribution of 11.6m."

- Figure 1 - Perhaps it would be better to show velocity with a logarithmic color scale? Velocity varies by several orders of magnitude over the ice sheet. Presenting it on a linear scale as in this figure has the effect of emphasizing fast-flowing areas, especially the ice shelf. Perhaps this was the intention; however, a side effect of this is that the structure of ice streams in the grounded part of

the domain is faded and difficult to see in many places. In addition, it would be helpful to show some aspects of the ice sheet geometry in additional plots of Figure 1, such as the bed elevation, ice thickness, or surface slope.

A logarithmic color scale was decided not to be added. Instead, we will change the colour scale of Figure 1 to highlight the ice streams more clearly. In addition, it is important to highlight that the front of the ice shelf is dynamic (i.e high velocity rates) in relation to the discussion in this paper about 'passive' part of the ice shelf. However, an additional figure will be added to Figure 1 to show the ice surface thickness (in m) to provide more context regarding the ice sheet geometry. These changes can be found in Figure 5 of this document.

- L68: "...ice viscosity depending on the ice temperature..." Where do you get the ice temperature from?

These sentences will be added to methods: "We use the ice temperature from ISSM's submission to the Ice Sheet Model Intercomparison Project for CMIP6 (ISMIP6) to initialize the ice viscosity over floating ice."

- L72-83: Model setup: What does your inverted drag coefficient look like? What about the other forcing fields? It would be useful to have a figure showing maps of the background fields used to force the model, so that we can put the sensitivity maps in context. I would be interested in seeing a figure that showed the inverted basal drag coefficient, the actual basal drag, the ice rigidity, surface mass balance, and basal mass balance. Without being able to see these fields, it is hard to put the sensitivity maps in context. Remember, the quantity being computed by the AD procedure is the linear sensitivity- that is, the derivative of the output quantity with respect to small perturbations in the input quantity, evaluated at the given value of the input quantity. If the background ice sheet configuration changes, the sensitivity will change as well.

A figure showing the inverted basal drag coefficient, actual basal drag, the ice rigidity, surface mass balance and basal mass balance will be included in the Appendix of the final manuscript (Figures 3 and 6 in this document). We can see that for the basal drag coefficient and basal drag that the floating ice has low basal drag (as expected), while the grounded ice shows variations in basal drag especially at the Siple Coast Ice Streams grounding zones. We can also see high basal drag at the pinning points and shear zones (i.e Ross Island). For ice viscosity we see high viscosity values for the floating ice (as expected) which decreases towards the grounded ice. For the basal mass balance we observed high basal melt at the ice shelf calving front, in particular at Coulman High/Ross Island. Relatively high basal melting is also observed towards the grounded ince. Finally, SMB has relatively constant values on the grounded and floating ice with the ice shelf calving front, coastal areas and transantarctic mountains being the exception to this.

- L102: Equation 4, compared to Figures 2 and 5 This equation implies that you have the units wrong in the labels of Figures 2 and 5. In order to get from

the sensitivity to a parameter, and the perturbation in that parameter, to an estimate in the volume change, it is necessary to perform a spatial integral over the model domain, with differential. Thus, the sensitivities should have units of meters per (parameter units), rather than m 3 per (parameter units), as you have given in the figure titles.

Thank you for pointing this out, we will change the units to units of meters per (parameter units).

- L107: "These sensitivity maps show that the vast majority of the grounded ice is not sensitive to changes in friction or rheology..." This wording isn't quite right. Presumably, the local flow in the vast majority of the grounded ice is going to be sensitive to local changes in friction or rheology. What you mean here is what you explained in the rest of the sentence, namely, that the overall mass balance of the ice sheet (after 20 years) is not sensitive to local changes in friction or rheology in the vast majority of the grounded domain. Perhaps a better wording would be, "These maps show that the sensitivity to friction and rheology is low in the vast majority of the grounded domain..."

The sentence will be changed to: "These maps show that the sensitivity to friction and rheology is low in the majority of the grounded domain, which means that changing the basal friction or ice rigidity over the majority of the region would not significantly influence the overall mass balance over 20 years."

- L118-120: "Kamb Ice Stream on the Siple Coast shows low to no sensitivity to changes in the basal friction or ice rigidity highlighting that the Kamb Ice Stream is currently stagnant and thus changing the friction or ice rigidity will not change the ice discharge significantly." It is worth pointing out that this result is a byproduct of the assumptions inherent in a linear perturbation analysis, and thus this negative result for Kamb ice stream can potentially be misleading. The sensitivity maps produced by this method basically answer the question: 'if we changed the friction coefficient at this particular location by an infinitessimal amount, what would be the marginal change in VAF?' In the continuous limit, each perturbation would only be to a single point with vanishing area, while in the numerical implementation, the perturbation resides at a single mesh node. Thus, the method cannot capture the sensitivity of VAF to a general reactivation of Kamb ice stream, which would involve a spatially correlated large-amplitude reduction in basal friction across the entire bed of the former ice stream. Clearly, a general reactivation of Kamb ice stream would reduce the grounded volume of the ice sheet. However, a reduction in basal friction at one location within the former ice stream would not have much effect if friction remained high across the rest of Kamb. Thus, the result of a low sensitivity within Kamb Ice Stream is accurate within the assumptions of this method, but it can be misleading in the sense that this method doesn't test for the possibility of spatially correlated perturbations to model parameters, and a general ice stream reactivation would be a spatially correlated perturbation. As I mentioned above, this method tests the sensitivity of VAF to perturbations in the forcing parameters, assuming that the general ice sheet configuration does not change. The reactivation of a dormant ice stream is a big enough perturbation that it would constitute a change in the general ice sheet configuration.

Clarification will be added to L118-120: "Kamb Ice Stream on the Siple Coast shows low to no sensitivity to changes in the basal friction or ice rigidity, highlighting that the Kamb Ice Stream is currently stagnant and thus changing the friction or ice rigidity will not change the ice discharge significantly. The sensitivity map tests the sensitivity of the VAF to perturbations in the forcing parameters, assuming that the general ice sheet configuration does not change. Therefore, the sensitivity map is unable to capture the general reactivation of the Kamb Ice Stream as the reactivation of a dormant ice stream would be a large perturbation that would constitute a change in the general ice sheet configuration."

- Figure 3: Are the negative values in the bottom row (sensitivity to rheology B) real, or do these simply represent numerical artifacts? Put another way, do you actually believe that there are places where strengthening the ice will actually cause more of it to flow into the ocean? There are also some negative values in the top row of this figure and in figure 4, but those negative excursions are much smaller than the negative excursions in the bottom row of figure 3.

The high spikes found in Figure 3 and 4 are numerical artefacts as our numerical model's treatment of the grounding line follows a sub-element grid scheme (i.e. subelement paramterization). Morlighem et al 2021 also found this and suggested that any sensitivity within 1-element of the grounding line should be interpreted with caution. This will be added to our results.

- Figure 5, L127-149, L192-201: Sensitivity to SMB and discussion of sensitivity to SMB I would be very interested in seeing a map of the sensitivity to SMB normalized by the stagnant ice sensitivity- that is, the sensitivity that you would get assuming that the ice did not move and the perturbation to SMB simply piled up mass at the location of the perturbation. That "stagnant ice" sensitivity is simply given by the time period of the simulation, in this case 20 years (by the way, I believe that you have made an additional units error in Figure 5, beyond the units error I pointed out earlier: the sensitivities to surface and basal mass balance probably should have units of m/(m/s), rather than m/(m/a) as you put in Figure 5, since otherwise values on the order of 6-7x10 8 are far too large. However, if ISSM is producing sensitivities in terms of m/s instead of m/a, then those magnitudes are exactly what one would expect from the stagnant-ice sensitivity with a 20 year model runtime). Normalizing the SMB sensitivity by the stagnant-ice sensitivity should help to put the SMB sensitivity in context. If there are any areas with sensitivity greater than the stagnant-ice value, then you could say definitively that SMB is vital to keeping those areas grounded and buttressing the rest of the ice sheet. Areas below the stagnant-ice value probably reflect regions where ice flux evolves very quickly (within the 20 year runtime) to export additional mass.

We will change the sensitivity maps (Figure 5 in this document) to show sensitivities to surface and basal mass balance with units of m/(m/a). Additionally, a 40 year transient simulation was ran to compare results with the 20 year transient simulation (Figure 1 of this document). These results were qualitatively similar suggesting that the areas highlighted in the sensitivity maps are robust. Extending the length of the transient simulation multiplies the gradient of sensitivities but the sensitivity patterns remain quantitatively similar to the short transient simulation. This was also found in Morlighem et al 2021. The 40 year transient simulation results will be included as the final results in the revised manuscript. Sensitivity maps to SMB normalised to stagnant ice sensitivity are included in Figure 8 in this document. However, we do not think this figure should be included in the final manuscript as it does not bring additional value to the discussion and the readers may find it difficult to grasp. We think that the sensitivity maps already included in the manuscript are appropriate for this paper and allow reliable conclusions to be drawn. In addition, Morlighem et al 2021 also used the same sensitivity maps allowing comparison between these two model domains sensitivity maps. We feel this would be an unnecessary additional plot that would not provide substantial analysis or differing conclusions to be drawn regarding the sensitivity analysis of SMB forcing on the model domain. In particular normalising the SMB sensitivities to stagnant ice flow, we feel would not be appropriate to include as we are not looking at sensitivities in relation to ice flow dynamics. In addition, we have shown that extending the length of the transient simulation multiplies the gradient of sensitivities but the sensitivity patterns remain the same for the SMB forcing in the model domain. We also showed that using the Budd or Weertman sliding law produced similar sensitivities to SMB forcing in the model domain. Therefore, this additional plot would not be included in the final manuscript.

- L157-158: "Our results show that the Bindschadler and MacAyeal Ice Streams have high sensitivities to changes in basal friction at their grounding zones due to these 'sticky spots' being key in controlling the ice discharge" Do your results in fact show high sensitivity at the sticky spots, as opposed to high sensitivity in the ice plains generally? This is where it would be helpful to have a map of the inversion results to compare your sensitivity maps to. In addition, this is where it would be helpful to distinguish between the sensitivity to perturbations in drag and the sensitivity to relative perturbations in drag, as I discussed earlier.

We will add the inversion results and Weertman friction law sensitivity maps to the Appendix of the final manuscript (Figures 2 and 3 in this document). With closer inspection of these inversion results and comparison to the sensitivity maps we still suggest that the sticky spots play an important role in the ice discharge. Wording on L157-158 will be clarified and edited.

**References**

- Dinniman, M. S., J. M. Klinck, E. E. Hofmann, and W. O. Smith, Effects of projected changes in wind, atmospheric temperature, and freshwater inflow on the ross sea, *Journal of Climate*, 31(4), 1619 – 1635, doi:10.1175/JCLI-D-17-0351.1, 2018.
- Morlighem, M., D. Goldberg, T. Dias dos Santos, J. Lee, and M. Sagebaum, Mapping the sensitivity of the amundsen sea embayment to changes in external forcings using automatic differentiation, *Geophysical Research Let*ters, 48(23), e2021GL095,440, doi:https://doi.org/10.1029/2021GL095440, e2021GL095440 2021GL095440, 2021.
- Smith Jr., W. O., M. S. Dinniman, E. E. Hofmann, and J. M. Klinck, The effects of changing winds and temperatures on the oceanography of the ross sea in the 21st century, *Geophysical Research Letters*, 41(5), 1624–1631, doi: https://doi.org/10.1002/2014GL059311, 2014.
- Stewart, C. L., P. Christoffersen, K. W. Nicholls, M. J. Williams, and J. A. Dowdeswell, Basal melting of Ross Ice Shelf from solar heat absorption in an ice-front polynya, *Nature Geoscience*, 12(6), 435–440, doi:10.1038/s41561-019-0356-0, 2019.

Figure 1: Sensitivity maps of final volume above flotation to the basal friction coefficient  $C_b$  (left) and ice rigidity B (right) over 40 years using the Budd linear sliding law are shown in the top row if the figure. The blue lines highlight the tracks for the along-flow profiles and the red lines the across-flow profiles. Sensitivity maps of final volume above flotation to the surface mass balance  $\dot{M}_s$  (left) and basal melting  $\dot{M}_b$  (right) over 40 years using the Budd linear sliding law are shown in the bottom row of the figure.

---

## Referee Report (RR1)

**Second Review of "Sensitivity of the Ross Ice Shelf to environmental and glaciological controls", by Baldacchino et al.**

**Major Comment**

In the first round of reviews for this manuscript, I expressed a concern about the use of a linear sliding law, and I suggested that the authors needed to perform additional analysis to demonstrate that their results are robust to the choice of sliding law prior to publication. The authors responded to my concern by repeating their experiment with a Weertman sliding law in addition to the Budd law that they originally used. I appreciate the effort that the authors have gone through to run an additional set of sensitivity experiments, which is more than the extra analysis that I requested originally. However, the new experiment doesn't actually address my concern.

My concern was specifically with respect to the nonlinearity of the sliding law. That is, I was concerned that they were using a value of the sliding exponent of m=1. Perhaps I was not clear enough in my original review. If that is the case, I apologize. Thus, I would like to emphasize here what exactly I was concerned about.

In their original experiment, the authors used a Budd sliding law, which has the following form:

$$\tau_b = C_b N v_b^{1/m} \qquad\qquad\qquad\qquad \text{eq. 1}$$

where $\tau_b$ represents the basal shear stress, $C_b$ represents the drag coefficient, N represents effective pressure, $v_b$ represents the basal shear stress, and m represents the slip exponent. A Weertman law is identical to a Budd law except that it omits any dependence on effective pressure, like so:

$$\tau_b = C_b v_b^{1/m} \quad . \qquad\qquad\qquad\qquad \text{eq. 2}$$

However, the Budd law and the Weertman law are identical in the sense that they both contain a power-law relationship between basal drag and basal sliding velocity. In the authors' original experiment, they set m=1, creating a linear relationship between slip and drag. However, there is a large body of work, including observational evidence, theoretical derivations, and numerical modeling results, indicating that a linear relationship between slip and drag is unrealistic. I cited a few papers making this case in my original review.

Thus, replacing the Budd law with a Weertman law does not, in itself, address my concern. My concern was that, in their original paper, the authors used a value of m=1 for the exponent in their sliding law, in contradiction to well-founded work which indicates that the relationship between slip velocity and shear stress ought to be highly nonlinear. Replacing a linear Budd law with a linear Weertman law does nothing to address this concern. What I wanted was some exploration of whether their results and interpretation would be robust if the linear sliding law they used was replaced with a nonlinear one.

To be fair, the authors do not state what value of the slip exponent they used for their Weertman experiment. The units of subplot (a) in figure A3 imply that they used a value of m=3, which would satisfy my concern as it means that they have tried a value of the exponent other than 1. If that is in fact the case, then all the authors need to do is to explicitly state what value of the slip exponent they used in the Weertman law, and my concern would be satisfied.

However, I cannot be sure what value of the slip exponent the authors have used simply from the units in the plot label, as the units could be mislabeled (as I mention in a minor comment below, the units are also mislabeled in other subplots of that figure). If, however, the authors have used a linear Weertman sliding law, then I must reluctantly insist that they have **not** addressed my major concern from the first round of review. When I expressed concern about the sensitivity of their results to the choice of sliding law, I was specifically concerned about the fact that they used a

linear relationship between basal shear stress and basal sliding velocity. Simply replacing a linear Budd law with a linear Weertman law does not address this concern.

If it is the case that the new sensitivity test used a linear Weertman sliding law, then I would suggest one of the following two strategies to address my concern: 1) since the authors are willing to perform additional sensitivity experiments, they could perform a test using a nonlinear Weertman or nonlinear Budd law, with a slip exponent of at least m=3; or, 2) they could analyze their existing results in the manner I suggested in the first review, and present the results of that analysis in additional supplemental figures.

**Minor Comments**

Figure 1:
"ice surface thickness"
"Ice surface thickness" sounds weird. After all, the thickness is computed from both the surface and the base, not the surface alone. Just plain "ice thickness" makes more sense.
"polarstereographic" → "polar sterographic"
In addition, the formatting of the degree symbol in -71º needs to be fixed.

L70-73: "The basal friction is based on a Budd friction law (Budd et al., 1979), in which basal drag is directly proportional to sliding velocity. This friction law may not be valid under some sectors of ourmodel domain such as the Siple Coast. Therefore, we performed additional experiments to test the sensitivity of our results to the Budd friction law (Figure A3) by using a Weertman friction law instead."
As I stated in my major comment, this is where you need to state what value of the sliding exponent you used in the Weertman law. Replacing a linear Budd law with a linear Weertman law does not change the fact that "basal drag is directly proportional to sliding velocity". If you did, in fact, use a nonlinear Weertman law, then my major concern could be satisfied by simply stating the value of the slip exponent here.

Figure 5:
I believe that the units are wrong in the caption. Sensitivity should be m/(parameter units), which in the case of surface and basal mass balance would be m/(m/s).

Section 4.4 Limitations
Again, the Weertman law does not really present anything independent of the Budd law if the Weertman law also used a linear relationship between basal stress and basal slip. If the Weertman law used a value of the exponent other than m=1 then that fact needs to be stated here.

Figure A1.
Thanks for including these maps, they are helpful for putting the results in context.

Figure A2.
Thanks for including this figure as well. However, I think you should double-check the units of the plot. You have a color scale from 0 to 1000, with no units labeled. If this is supposed to be units of Pa, then the maximum is way too low, but if this is supposed to be kPa, then the maximum is way too high. The spatial pattern in the map looks reasonable but you should really double-check the units and the magnitudes here.

Figure A3.
The units label for plot (a) implies that you used a value of m=3 in the Weertman law, which would satisfy my major concern described above. However, I can't be sure that you have labeled them

correctly, because the units labels for plots (c) and (d) are wrong. Those labels should be m/(m/s) rather than m/(m/s)$^{1/3}$.

---

## Referee Report (RR2)

**Sensitivity of the Ross Ice Shelf to environmental and glaciological controls**

**by Baldacchino et al.**

**-Second revision-**

I want to thank the authors for having addressed most of my comments during their revisions. I found their answers to my questions very informative and detailed. I also appreciate the extra work, pushing the time of simulation to 40 years as well as the use of a second (non-linear?) friction law as suggested by the other reviewer. The restructuration of the discussion, e.g., less back and forth between "basal friction" and "ice rigidity", as well as the addition of subsections really makes the discussion more reader-friendly.

I have very few additional comments to make (see after), mostly asking for a few precisions that might help the reader. The most obvious, which has nothing to do with the quality of the scientific work, is the number of typos and mispunctuations that, I think, need to be corrected before publication. I have pointed out some of these in my technical comments but I am sure I missed others.

In conclusion, I believe that this work, corroborating the main results of Morlighem et al. (2021) but on a different region, will be useful to the community. It is therefore well suited for publication in The Cryosphere, after very minor revisions.

**Specific comments:**

- You ran additional simulations with a Weertman Friction law ($\tau_b = C|\boldsymbol{u}_b|^{m-1}\boldsymbol{u}_b$) but you did not mention what exponent (m) you used in the law, I suppose that $m \neq 1$, since the goal was to introduce a non-linear relationship between the velocity and the basal drag. If the exponent is $m = 1$ (which I doubt), then I think that the extra results might not be that useful to the manuscript and that a non-linear friction law should be used instead. Could you mention the exponent you use and maybe add the equation in the Appendix? A proper reference to Weertman is also missing, probably "*Weertman, On the Sliding of Glaciers, JoG, 1957*".
- I am sorry if I mention this idea only now but I think that some numbering such as "Figure Xa,b,c,…" and associated references in the text would really help navigating between the text and the figures. Keeping the name is fine but additional "lettering" would nice. For example, line 218 could write: "*Our results show that Bindschadler Ice Stream has the highest sensitivity to changes in ice rigidity at the margins (Fig. 4e), while MacAyeal Ice Stream has a higher sensitivity in the main ice trunk (Fig. 4f).*"
- **Line 198:** the basal friction within the main trunk is more important than what? More important than in the shear margins? I think this statement is really clear for Binschadler Ice Stream but not that much for MacAyeal Ice Stream. At least this is what I see when comparing Figure 4b and 4c.
- **line 220:** I think I did not catch this one during my first review. "*Meyer and Minchew (2018) show that Bindschadler Stream has temperate zones of ice within its shear margins and*

*thus changes in ice rigidity here would influence the ice stream discharge as shown by our results."* Could you explain why a change in temperate zones is any different to a change in a cold zone? I think that what matters is the intensity of the change in rigidity, not the absolute value of the rigidity.

- **4.4 Limitations**: Can you add a unit for the basal drag coefficient (and maybe specify that this coefficient is the one for Budd law you used during the inversion)?
- You use alternatively "yr" (e.g., fig A1) and 'a' for years (e.g., Fig 2 or Fig A3). Could you use one or the other only? You give the units for sensitivity maps (Fig 2 and 5) but not for the grounding-line and along-flow profiles (Fig 3 and 4). Could you add those?

**Technical comments and typos:**

- **Line 50:** Thank you for the edition of this sentence. Please, just add some punctuations to the sentence, e.g., *"(Dinniman et al., 2018), which will highly likely increase ice-shelf basal melting and, subsequently, the future stability of the RIS (Stewart et al., 2019)."*
- **Line 62:** I am not sure I understand the correction, i.e., keeping only *"[…] which changes in external forcings and internal material properties of the ice effect the overall mass balance […]".* Do you mean "affect" instead of "effect"?
- **Line 79:** I'd suggest to rewrite *"(i.e. Sub-element Parameterization 1 in Seroussi et al., 2014)"* .
- **Line 149:** I still think that passive ice should be introduced in the text and not only in Figure 5.
- **Line 188:** leading to *"leading to an thickening"*
- **Line 193:** delete the last part, since basal friction is always at the bed, i.e., *"by the basal friction conditions at the bed."*
- **Line 205:** *"Our results show that Whillans Ice Stream is highly sensitive to changes in basal friction at its shear margins, suggesting that changes in lubrication conditions here influences the flow and discharge rates of the ice stream."*
- **Line 212:** To me, this sentence is almost a repetition of the previous paragraph, I don't understand why you introduce a new paragraph here.
- **Line 240**: *"[…] and this makes it highly sensitive to changes in ice rigidity, which is also shown by our results".*
- **Line 324:** I'd change "but the sensitivity patterns remain quantitatively qualitatively similar" since the "quantity" doubles when you double the simulation period.
- **Figure A3:** "Top row if of the figure"

---

## Author Response (AR2)

**Sensitivity of the Ross Ice Shelf to environmental and glaciological controls -Response to reviewers-**

Francesca BALDACCHINO et al

August 22 2022

We would like to thank the two anonymous reviewers and the editor for their further positive and constructive comments. We address their remarks below point by point.

**1 Reviewer #1**

**1.1 Major Comments**

- In the first round of reviews for this manuscript, I expressed a concern about the use of a linear sliding law, and I suggested that the authors needed to perform additional analysis to demonstrate that their results are robust to the choice of sliding law prior to publication. The authors responded to my concern by repeating their experiment with a Weertman sliding law in addition to the Budd law that they originally used. I appreciate the effort that the authors have gone through to run an additional set of sensitivity experiments, which is more than the extra analysis that I requested originally. However, the new experiment doesn't actually address my concern. My concern was specifically with respect to the nonlinearity of the sliding law. That is, I was concerned that they were using a value of the sliding exponent of m=1.

The slip exponent used in this manuscript for the Weertman experiment was a value of m = 3. We are sorry for not stating this more clearly in the revised manuscript. Therefore, we satisfy your concern regarding the use of the sliding exponent of m = 1. This has been clarified in the methodology and limitations section of the manuscript.

**1.2 Minor Comments**

- Figure 1: replace 'ice surface thickness' with 'ice thickness'. Change 'polarstereographic" → "polar sterographic". In addition, the formatting of the degree symbol in -71º needs to be fixed

Done.

- L70-73: "The basal friction is based on a Budd friction law (Budd et al., 1979), in which basal drag is directly proportional to sliding velocity. This friction law may not be valid under some sectors of our model domain such as the Siple Coast. Therefore, we performed additional experiments to test the sensitivity of our results to the Budd friction law (Figure A3) by using a Weertman friction law instead". As I stated in my major comment, this is where you need to state what value of the sliding exponent you used in the Weertman law. Replacing a linear Budd law with a linear Weertman law does not change the fact that "basal drag is directly proportional to sliding velocity". If you did, in fact, use a nonlinear Weertman law, then my major concern could be satisfied by simply stating the value of the slip exponent here.

The slip exponent for the Weertman Sliding law of m = 3 has been stated here.

- Figure 5: I believe that the units are wrong in the caption. Sensitivity should be m/(parameter units), which in the case of surface and basal mass balance would be m/(m/s).

Units have been changed to m/(m/s).

- Again, the Weertman law does not really present anything independent of the Budd law if the Weertman law also used a linear relationship between basal stress and basal slip. If the Weertman law used a value of the exponent other than m=1 then that fact needs to be stated here.

The Weertman Sliding law used an exponent of m = 3 and this has been clarified in this section.

- Figure A1. Thanks for including these maps, they are helpful for putting the results in context.

No problem, thanks for suggesting them!

- Figure A2. Thanks for including this figure as well. However, I think you should double-check the units of the plot. You have a color scale from 0 to 1000, with no units labeled. If this is supposed to be units of Pa, then the maximum is way too low, but if this is supposed to be kPa, then the maximum is way too high. The spatial pattern in the map looks reasonable but you should really double-check the units and the magnitudes here.

The units are in kPa and the magnitude has been checked (now 0 to 200 kPa). The figure has been replotted. Thank you for pointing this out!

- Figure A3. The units label for plot (a) implies that you used a value of m=3 in the Weertman law, which would satisfy my major concern described above. However, I can't be sure that you have labeled them correctly, because the units labels for plots (c) and (d) are wrong. Those labels should be m/(m/s).

The units have been changed for plots (c) and (d).

[Figure]

Figure 1: Inverted basal drag ($kPa$).

**2 Reviewer #2**

**2.1 Specific Comments**

- You ran additional simulations with a Weertman Friction law but you did not mention what exponent (m) you used in the law, I suppose that m is not 1, since the goal was to introduce a non-linear relationship between the velocity and the basal drag. If the exponent is m = 1 (which I doubt), then I think that the extra results might not be that useful to the manuscript and that a non-linear friction law should be used instead. Could you mention the exponent you use and maybe add the equation in the Appendix? A proper reference to Weertman is also missing.

The Weertman Friction law used exponent of m = 3 for these additional simulations. This has now been clarified throughout the manuscript in the methodology and limitations sections. A proper reference to Weertman has also now been included *Weertman* [1957]. And the equation for the Weertman Friction law has been included in the limitations section.

- I am sorry if I mention this idea only now but I think that some numbering such as "Figure Xa,b,c,..." and associated references in the text would really help navigating between the text and the figures. Keeping the name is fine but additional "lettering" would nice. For example, line 218 could write: "Our results show that Bindschadler Ice Stream has the highest sensitivity to changes in ice rigidity at the margins (Fig. 4e), while MacAyeal Ice Stream has a higher sensitivity in the main ice trunk (Fig. 4f)."

This has been edited throughout the manuscript to help the reader more easily navigate between the text and figures. Thank you for this suggestion!

- Line 198: the basal friction within the main trunk is more important than what? More important than in the shear margins? I think this statement is really clear for Binschadler Ice Stream but not that much for MacAyeal Ice Stream. At least this is what I see when comparing Figure 4b and 4c

This has now been rephrased and the word 'more' has been removed. We still think that this statement fits in with the MacAyeal Ice Stream as we can see that there is high sensitivity within the main ice trunk.

- line 220: I think I did not catch this one during my first review. "Meyer and Minchew (2018) show that Bindschadler Stream has temperate zones of ice within its shear margins and thus changes in ice rigidity here would influence the ice stream discharge as shown by our results." Could you explain why a change in temperate zones is any different to a change in a cold zone? I think that what matters is the intensity of the change in rigidity, not the absolute value of the rigidity.

This line has been rephrased to explain the importance of the temperate ice in influencing changes in ice rigidity more clearly: "*Meyer and Minchew* [2018] show that Bindschadler Ice Stream has temperate zones of ice within its shear margins, with these zones being at melting temperature and controlling the rate at which the ice stream flows by softening the ice rigidity. Therefore, changes in the ice rigidity of the Bindschalder Ice Stream margins would influence the ice stream discharge as shown by our results."

- 4.4 Limitations: Can you add a unit for the basal drag coefficient (and maybe specify that this coefficient is the one for Budd law you used during the inversion)?

The unit has been added in the limitations section and clarification that the Budd linear sliding law was used during inversion has been added in the caption of Figure A1.

- You use alternatively "yr" (e.g., fig A1) and 'a' for years (e.g., Fig 2 or Fig A3). Could you use one or the other only? You give the units for sensitivity maps (Fig 2 and 5) but not for the grounding-line and along-flow profiles (Fig 3 and 4). Could you add those?

"yr" was removed from Figure A1 and replaced with 'a' to keep it consistent throughout the manuscript. The units for the grounding line and along-flow profiles (Figures 3, 4 and 6) were included in the figure captions. We did not include these units on the y axis labels as it resulted in the figure panels becoming overcrowded and less readable.

**2.2  Technical Comments and Typos**

- Line 50: Thank you for the edition of this sentence. Please, just add some punctuations to the sentence, e.g., "(Dinniman et al., 2018), which will highly likely increase ice-shelf basal melting and, subsequently, the future stability of the RIS (Stewart et al., 2019)."

Done.

- Line 62: I am not sure I understand the correction, i.e., keeping only "[...] which changes in external forcings and internal material properties of the ice effect the overall mass balance [...]". Do you mean "affect" instead of "effect"?

Yes, this has been changed from 'effect' to 'affect'.

- Line 79: I'd suggest to rewrite "(i.e. Sub-element Parameterization 1 in Seroussi et al., 2014)"

Done.

- Line 149: I still think that passive ice should be introduced in the text and not only in Figure 5.

This has been included in lines 163-164.

- Line 188: leading to "leading to an thickening"

Done.

- Line 193: delete the last part, since basal friction is always at the bed, i.e., "by the basal friction conditions at the bed."

Done.

- Line 205: "Our results show that Whillans Ice Stream is highly sensitive to changes in basal friction at its shear margins, suggesting that changes in lubrication conditions here influences the flow and discharge rates of the ice stream."

Done.

- Line 212: To me, this sentence is almost a repetition of the previous paragraph, I don't understand why you introduce a new paragraph here.

A new paragraph has been introduced here as we are discussing a different region of the model domain, Byrd Glacier, while the paragraphs above were discussing the Siple Coast Ice Streams.

- Line 240: "[...] and this makes it highly sensitive to changes in ice rigidity, which is also shown by our results."

Done.

- Line 324: I'd change "but the sensitivity patterns remain qualitatively similar" since the "quantity" doubles when you double the simulation period.

Done.

- Figure A3: "Top row of the figure"

Done.

**References**

Meyer, C. R., and B. M. Minchew, Temperate ice in the shear margins of the Antarctic Ice Sheet: Controlling processes and preliminary locations, *Earth and Planetary Science Letters*, *498*, 17–26, doi:10.1016/j.epsl.2018.06.028, 2018.

Weertman, J., On the sliding of glaciers, *Journal of Glaciology*, *3*(21), 33–38, doi:10.3189/S0022143000024709, 1957.